# Revisiting Residual Connections: Orthogonal Updates for Stable and Efficient Deep Networks

**Giyeong Oh**$^\diamond$ **Woohyun Cho**$^\diamond$ **Siyeol Kim**$^\diamond$ **Suhwan Choi**$^\heartsuit$ **Youngjae Yu**$^{\spadesuit *}$

**Yonsei University**$^\diamond$                    **Maum.AI**$^\heartsuit$
{hard2251,k106419,cykim0528}@yonsei.ac.kr          claude@maum.ai

**Seoul National University**$^\spadesuit$
youngjaeyu@snu.ac.kr

## Abstract

Residual connections are pivotal for deep neural networks, enabling greater depth by mitigating vanishing gradients. However, in standard residual updates, the module's output is directly added to the input stream. This can lead to updates that predominantly reinforce or modulate the existing stream direction, potentially underutilizing the module's capacity for learning entirely novel features. In this work, we introduce *Orthogonal Residual Update*: we decompose the module's output relative to the input stream and add only the component orthogonal to this stream. This design aims to guide modules to contribute primarily new representational directions, fostering richer feature learning while promoting more efficient training. We demonstrate that our orthogonal update strategy improves generalization accuracy and training stability across diverse architectures (ResNetV2, Vision Transformers) and datasets (CIFARs, TinyImageNet, ImageNet-1k), achieving, for instance, a +3.78 pp Acc@1 gain for ViT-B on ImageNet-1k. Code and models are available at https://github.com/BootsofLagrangian/ortho-residual.

## 1 Introduction

Residual connections [1] have been a cornerstone in deep learning, fundamentally enabling the training of substantially deeper neural networks by mitigating vanishing gradients. The original ResNet architecture [1] updated an internal state $x_n$ via $x_{n+1} = \sigma_{\text{act}}(x_n + f(x_n))$, where the non-linear activation $\sigma_{\text{act}}$ was applied after the summation, meaning $x_n$ did not propagate purely linearly. Subsequent work, notably ResNetV2 [2], introduced full linear mappings of the form $x_{n+1} = x_n + f(\sigma_{\text{pre}}(x_n))$. This design, where $\sigma_{\text{pre}}$ (e.g., normalization and activation) precedes the residual function $f$ and no transformation follows the addition, allows the unmodified $x_n$ to serve as a *linear residual stream* [3] that passes representation directly across layers. This principle of a linear residual stream is now a prevalent feature in many modern high-capacity architectures, including contemporary Transformers [4, 5] and large language models [6–12] that predominantly employ pre-layer normalization [2, 13].

In such architectures, complex modules $f$ (e.g., attention or MLP modules) operate on a (or normalized version of this, $\sigma(x_n)$) linear residual stream $x_n$, and their output $f(\sigma(x_n))$ is additively combined with $x_n$. Conceptually, this module output $f(\sigma(x_n))$ can be decomposed with respect to the input stream $x_n$ into two components: $f_\parallel$, parallel to $x_n$, and $f_\perp$, orthogonal to $x_n$. The

---

$^*$Corresponding author

39th Conference on Neural Information Processing Systems (NeurIPS 2025).

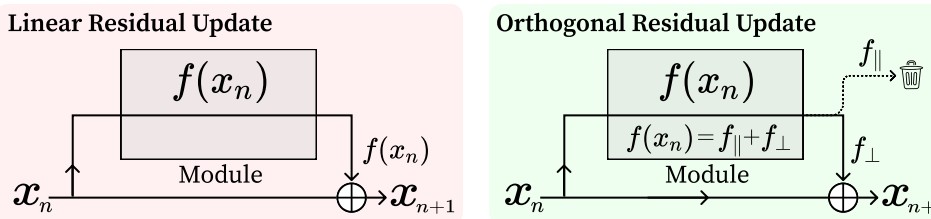

Figure 1: **Intuition behind our orthogonal residual update.** *Left:* The standard residual update adds the full output of module $f(x_n)$ to the input stream $x_n$. *Right:* Our proposed update first decomposes the module output $f(x_n)$ into a component parallel to $x_n$ ($f_\parallel$) and a component orthogonal to $x_n$ ($f_\perp$). We then discard $f_\parallel$ and add only the orthogonal component $f_\perp$ to the stream.

component parallel to $x_n$ can effectively rescale the stream, prompting the question of whether updates should prioritize directions not already present in $x_n$. These considerations raise whether module capacity is best allocated to novel directions rather than further modulation of the existing stream. Throughout, we write $f_\parallel = s_n x_n$ and $f_\perp = f(\sigma(x_n)) - s_n x_n$, where $s_n$ denotes the scalar projection of $f(\sigma(x_n))$ onto $x_n$.

Such scaling need not use the full expressivity of $f$; isolating $f_\perp$ emphasizes contributions in novel directions. The orthogonal component $f_\perp$, in contrast, inherently introduces new directional components into the representation space, distinct from the current stream $x_n$. We hypothesize that *by explicitly isolating and utilizing only the orthogonal component $f_\perp$ for updating the residual stream, modules are able to focus on contributing novel aspects to the representation.*

As a proof of concept, we study the *Orthogonal Residual Update*, replacing $x_{n+1} = x_n + f(\sigma(x_n))$ with $x_{n+1} = x_n + f_\perp(x_n)$. As conceptually illustrated in Fig. 1, instead of linear additive update $x_n + f(\sigma(x_n))$, our approach employs $x_n + f_\perp(x_n)$, where $f_\perp(x_n)$ is the component of the module's output $f(\sigma(x_n))$ explicitly made orthogonal to the input stream $x_n$. We evaluate this approach on ResNetV2 [2] and the Vision Transformer (ViT) [5] across standard image classification benchmarks, including the CIFAR datasets [14], TinyImageNet [15], and ImageNet-1k [16]. Across ResNetV2 and ViT on standard benchmarks, we observe improvements in generalization and distinct training dynamics, as shown in Figs. 2, 3. Our main contributions are:

- We re-examine additive updates in networks with linear residual streams, noting that the component parallel to the input stream may rescale—and at times become anti-aligned with—the stream, potentially impeding information propagation.
- We propose a simple, principled modification, *Orthogonal Residual Update*, which isolates and uses only the orthogonal component $f_\perp$ to encourage novel directions and mitigate interference from $f_\parallel$.
- We empirically show improvements in generalization, training stability, and overall efficiency across ViT and ResNetV2 on standard image classification datasets.

## 2 Related Works

**Modifying the Stream Itself** Beyond the standard linear skip connection [2], prior work has modified the skip path $I x_n$ to control propagation or encode inductive biases by replacing $I$ with fixed linear transforms $P$ or $\Gamma$ (e.g., norm-preserving orthogonal $P$ [17] or structure-inducing entangled $\Gamma$ [18]). These approaches fundamentally alter the skip (i.e., $I \to P/\Gamma$) and analyze how changed spectral/sparsity properties affect learning; in contrast, we preserve the linear skip $I$ and refine only the *additive update* $f(\sigma(x_n))$ combined with it.

**Orthogonality in Deep Learning** Orthogonality is a recurring principle in deep learning, valued for promoting stable signal propagation and beneficial representation properties. Orthogonal weight initialization [19, 20] and training-time enforcement via regularization or manifold-constrained optimization (e.g., Stiefel) [21–24] are common tools; more recently, Orthogonal Finetuning (OFT) adapts pre-trained models via orthogonal weight transforms [25]. At the optimizer level, Muon [26] orthogonalizes updates via Newton–Schulz iterations.

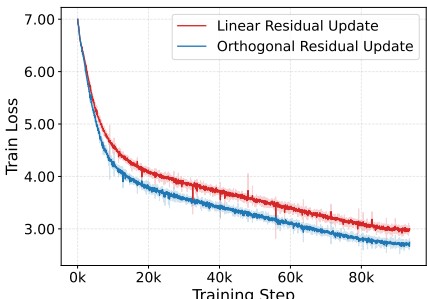 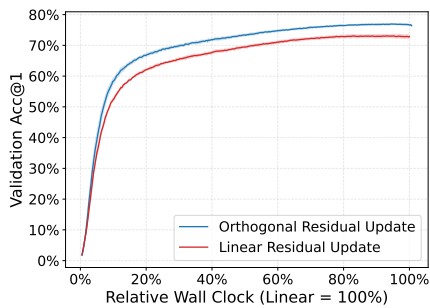

(a) Training loss vs. training iterations.   (b) Validation accuracy vs. relative wall-clock.

Figure 2: Orthogonal update accelerates convergence and enhances generalization efficiency compared to the standard linear update baseline (ViT-B on ImageNet-1k results shown). (a) **Faster Convergence:** Orthogonal update (blue) achieves significantly lower training loss in fewer iterations. (b) **Improved Time-to-Accuracy:** Orthogonal update (blue) attains higher validation Acc@1 consistently outperforms linear update with minimal overhead.

## 3 Orthogonal Residual Updates

### 3.1 Preliminary

For any non-zero vector $x_n \in \mathbb{R}^d$, a module output $f(\sigma(x_n))$ can be uniquely expressed as the sum of two distinct components: one component that is directly related to the current stream $x_n$, and another component that is independent of $x_n$. We can write this abstractly as:

$$f(\sigma(x_n)) = f_\parallel + f_\perp, \tag{1}$$

where $f_\parallel$ is the component parallel to $x_n$, and $f_\perp$ is the component orthogonal to $x_n$. Since $f_\parallel$ must lie along the direction of $x_n$, it necessarily takes the form $f_\parallel = \alpha x_n$. This projection step, analogous to the fundamental operation in the Gram-Schmidt orthogonalization process, yields $\alpha$ as follows:

$$f_\parallel = \alpha x_n, \text{ where } \alpha = \frac{\langle x_n, f(\sigma(x_n)) \rangle}{\langle x_n, x_n \rangle} \quad \text{and} \quad f_\perp = f(\sigma(x_n)) - f_\parallel. \tag{2}$$

Here $\langle \cdot, \cdot \rangle$ denotes the dot product. Consequently, the orthogonal component $f_\perp$, capturing the part of $f(\sigma(x_n))$ linearly independent from $x_n$, is obtained by subtracting this parallel component. By construction, this component satisfies $\langle x_n, f_\perp \rangle = 0$. Interpreting these components, $f_\parallel$ signifies the portion of the module's output that merely scales the direction already present in the stream $x_n$. Conversely, $f_\perp$ indicates the 'novel' representation contributed by the current module relative to $x_n$.

### 3.2 Orthogonal-only Update Rule

Building upon the decomposition introduced in the previous subsection, we now propose our core update mechanism. Instead of adding the full module output $f(\sigma(x_n))$ to the stream as in standard residual updates, we advocate for using only the component $f_\perp(x_n)$ derived from $f(\sigma(x_n))$, as illustrated in Fig. 1. This component is explicitly defined as:

$$f_\perp(x_n) = f(\sigma(x_n)) - s_n x_n, \qquad \text{where} \quad s_n = \frac{\langle x_n, f(\sigma(x_n)) \rangle}{\|x_n\|^2 + \epsilon}. \tag{3}$$

Note that we now explicitly include the small constant $\epsilon > 0$ in the denominator for numerical stability during computation, a detail omitted in the ideal decomposition in Sec. 3.1. Throughout our experiments, we set $\epsilon = 10^{-6}$; sensitivity is reported in Sec. 4.7.

Our proposed orthogonal-only update rule is thus simply:

$$\boxed{x_{n+1} = x_n + f_\perp(x_n).} \tag{4}$$

This formulation is motivated by two key properties. First, from a geometric perspective, it acts as an efficient approximation of an exponential map on the representation manifold, leveraging the local Euclidean structure of the tangent space at $x_n$. Second, from a practical optimization standpoint, it preserves the identity gradient path essential for stable training of deep networks. The full derivation is deferred to Appendix B.

## 3.3 Implementation and Computational Overhead

---

**Algorithm 1** Orthogonal Residual Update

---

**Input:** stream $x_n$, module output $f$, dimension set $\mathcal{D}$ (non-batch dims to reduce over), stability $\epsilon > 0$.

1:   $s_{\mathrm{m}} \leftarrow \mathtt{sum}(x_n \odot f, \mathcal{D}, \mathtt{keepdim=True})$
2:   $s_{\mathrm{d}} \leftarrow \mathtt{sum}(x_n \odot x_n, \mathcal{D}, \mathtt{keepdim=True})$
3:   $s_{\mathrm{n}} \leftarrow s_{\mathrm{m}}/(s_{\mathrm{d}} + \epsilon)$
4:   $f_\perp \leftarrow f - s_{\mathrm{n}} \odot x_n$

**return** $x_n + f_\perp$

---

Alg. 1 subsumes both variants via the choice of the reduction set $\mathcal{D}$: *feature-wise* uses $\mathcal{D} = \{\mathtt{idx\_f}\}$, i.e., reducing over the feature axis (Transformer: $d$; CNN: channel dimension $C$), whereas the *global* variant sets $\mathcal{D}$ to all non-batch dimensions (i.e., flatten–then–reduce). Setting $\mathcal{D}$ to all non-batch dimensions is also possible; however, it ignores structural priors of attention heads and convolutional kernels. Since global projection can interfere with attention, we only evaluate the global variant on CNNs in this paper.

(a) Approximate FLOPs per Transformer block. $s = n_{\mathrm{seq}}$, $d = d_{\mathrm{model}}$; FFN assumes a $4d$ expansion. Our *feature-wise* orthogonal connection introduces only $O(sd)$ FLOPs on top of the block.

| Module | Connection | Total FLOPs |
|---|---|---|
| Attention | Linear | $\approx 8sd^2 + 4s^2d + sd$ |
| | Orthogonal | $\approx 8sd^2 + 4s^2d + sd + \mathbf{6sd} + \mathbf{2s}$ |
| MLP (FFN) | Linear | $\approx 16sd^2 + sd$ |
| | Orthogonal | $\approx 16sd^2 + sd + \mathbf{6sd} + \mathbf{2s}$ |

(b) Training throughput (img/s) and overhead (%) of **Ortho-F** relative to the linear residual baseline.

| Arch. | Linear | Ortho-F | Overhead |
|---|---|---|---|
| ResNetV2-34 | 1737.2 | 1634.0 | 5.94% |
| ResNetV2-50 | 1002.8 | 876.7 | 12.58% |
| ViT-S | 3476.1 | 3466.3 | 0.28% |
| ViT-B | 1270.1 | 1246.2 | 1.88% |

Table 1: **Computation vs. practice.** Orthogonal projection adds $O(sd)$ FLOPs per block (bold in (a)); throughput in (b) is measured under identical conditions.

We instantiate the update either *feature-wise* (default) or *global*, both captured by Alg. 1 with different reduction sets $\mathcal{D}$. Feature-wise is vectorization-friendly and requires only $O(sd)$ FLOPs per block ($\approx 6sd$ for the projection plus $\approx 2s$ for normalization), negligible relative to attention/FFN (Tab. 1a). Empirically, the throughput overhead of **Ortho-F** (Orthogonal feature-wise) is modest (Tab. 1b): $\leq 2\%$ on ViT-S/B and $\sim$3–13% on ResNetV2 under identical batch sizes and hardware; detailed PyTorch code and AMP/compilation notes are in Appendix G.

## 3.4 Observed Internal Dynamics of Orthogonal Updates

Before turning to large-scale results, we first examine the *in-block dynamics* of ViT-S on TinyImageNet (5 seeds). We track two diagnostics that probe the mechanism: (i) the stream norm $\|x_n\|^2$ and (ii) the magnitude of the parallel component of the module output, $\|f_\|(x_n)\|^2$, recalling that $f_\|(x_n) = s_n x_n$ and $f_\perp(x_n) = f(\sigma(x_n)) - s_n x_n$ with $s_n$ defined in Eq. (3). Summary curves (MLP/Attention, blocks 0–5) are shown in Fig. 3; cosine similarity and additional per-block traces are deferred to Appendix D.

**The Transition Point.** We consistently observe an early, layer-dependent juncture where trends diverge (indicated by upward arrows in each panel): for the **linear** pathway, the parallel contribution diminishes and $\|x_n\|^2$ typically exhibits a peak followed by a decrease; for the **orthogonal** pathway, $\|x_n\|^2$ stabilizes while the orthogonal component is maintained. Sign-sensitive trajectories (e.g., directional alignment) and the Jacobian-based analysis are deferred to Appendix D and B.4.

**Key observation I: Linear induces parallel suppression.** Across layers, the **linear** residual update exhibits a *reduction in the contribution of the parallel component* of the module output. Concretely, in Fig. 3 (b,d) the parallel-component energy $\|f_\|(x_n)\|^2$ for the linear baseline (red) typically declines after the Transition Point and often settles near a low plateau—most clearly in deeper MLP and Attention blocks—indicating that the model progressively suppresses $f_\|$ during training. In contrast, our orthogonal variant (blue) maintains or grows $\|f_\|\|^2$ as the module learns on its full output while

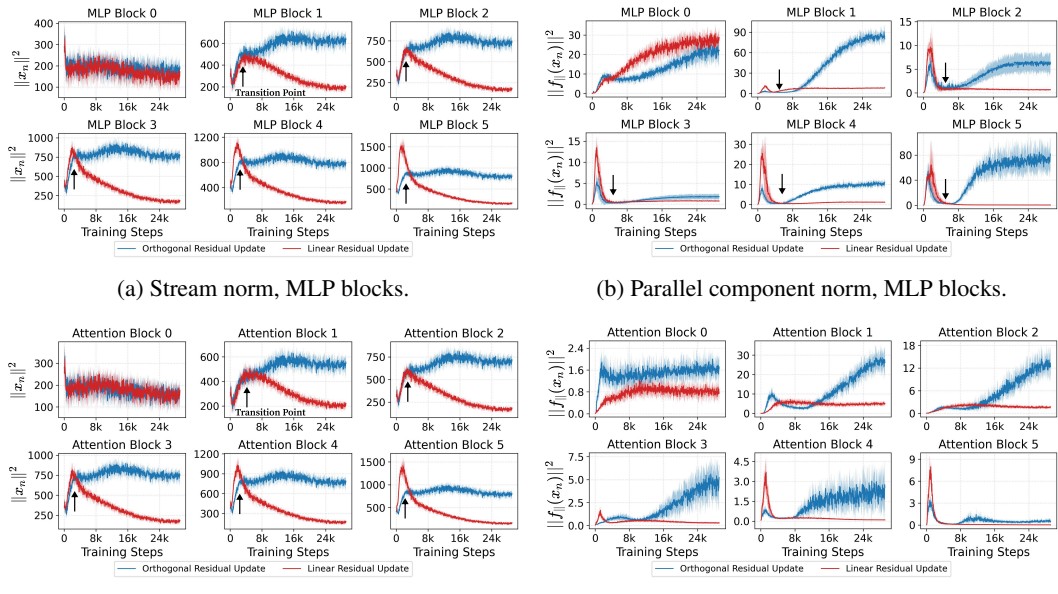

(a) Stream norm, MLP blocks.

(b) Parallel component norm, MLP blocks.

(c) Stream norm, Attention blocks.

(d) Parallel component norm, Attention blocks.

Figure 3: **Internal dynamics (ViT-S, TinyImageNet, 5 seeds).** Each subfigure shows blocks 0–5 (MLP top, Attention bottom). **Ours** denotes orthogonal updates; **Linear** denotes the standard residual. **(a,c)** After the *Transition Point*, orthogonal updates stabilize the stream norm $\|x_n\|^2$, whereas linear updates typically exhibit a post-transition decrease. **(b,d)** The parallel component energy $\|f_\|(x_n)\|^2$ follows distinct layer-wise profiles for linear vs. orthogonal updates. Signed parallel coefficients and orthogonal-component traces are analyzed in the Appendix D.

the *update* remains orthogonal. This confirms that parallel suppression occurs under the linear residual pathway.

**Key observation II: Orthogonal stabilizes the stream norm.** After the Transition Point (Fig. 3 (a), (c)), the **orthogonal** update $x_{n+1} = x_n + f_\perp(x_n)$ stabilizes the stream norm $\|x_n\|^2$ and the updates act predominantly as *rotations* on a near-constant–norm state. Importantly, we *do not* constrain the module $f(\cdot)$ itself to be orthogonal; we only add the orthogonal component of its output to the stream. *Thus $f(\cdot)$ may still contain a parallel part, and since $\langle x_n, f_\perp(x_n)\rangle \approx 0$ (finite $\epsilon$ and precision), parallel alignment can re-enter the module output*; see Appendix B for analysis. Although the module output $f(\sigma(x_n))$ can still *accumulate* a parallel component during learning (via gradients on $s_n$) and a small numerical bias may remain due to the stability constant $\epsilon$, the *update path* discards the parallel term by construction. Hence, there is no *direct* positive-feedback loop via the residual addition that would amplify or suppress the parallel component; instead, orthogonal energy is preserved and steers representation change directionally.

### 3.5 Stream Scaling vs. Geometric Projection

Having observed parallel suppression under the linear pathway, we ask whether simply learning a stream gain can reproduce this behavior. To test whether granting the model explicit freedom to rescale the stream leads it to suppress the parallel component, we consider a learned stream–scaling variant

$$x_{n+1} = x_n + \alpha_\ell\, x_n + f(x_n) \quad \text{(equivalently } x_{n+1} = (1 + \alpha_\ell)x_n + f(x_n)), \tag{5}$$

where $\alpha_\ell$ is a learnable per-layer scalar initialized at 0. Fig. 4 visualizes the trajectories $\alpha_\ell(\text{step})$ by block. Granting this freedom yields systematic suppression of the parallel pathway in MLP blocks (negative drift of $\alpha_\ell$) and small, near-zero or mildly positive values in Attention. Thus, learned rescaling primarily adjusts layerwise magnitude, whereas our method computes a stream-dependent projection $s_n(x_n)$ to remove the parallel component and update with $f_\perp(x_n)$. However, $\alpha_\ell$ is input-invariant: there is no single $\alpha_\ell$ that makes $\alpha_\ell\|x_n\|^2 + \langle f(\sigma(x_n)), x_n\rangle = 0$ for all $x_n$, so

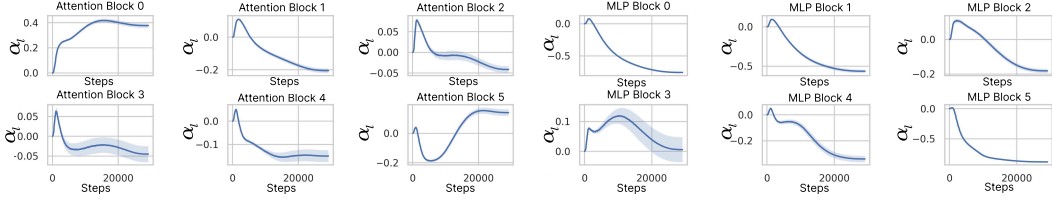

(a) Attention blocks: $\alpha_\ell(\text{step})$.      (b) MLP blocks: $\alpha_\ell(\text{step})$.

Figure 4: **Learned stream–scaling variant:** per-block trajectories of $\alpha_\ell$ (median across seeds; band = interquartile range). Values below zero correspond to stream attenuation. This diagnostic contrasts a layerwise, input-invariant scalar with our input-dependent projection.

simple scaling cannot *guarantee* an orthogonal update. In contrast, $s_n(x_n)$ depends on the current stream state and removes the parallel term. Beyond this learned rescaling, Appendix E introduces a unified residual family parameterized by $(\rho_\ell, \theta_\ell)$ that mixes parallel/orthogonal contributions, i.e., $x_{n+1} = x_n + \rho_\ell(\sin\theta_\ell\, f_\parallel(x_n) + \cos\theta_\ell\, f_\perp(x_n))$.

## 4 Experiments

### 4.1 Experimental Setting

**Datasets.** We evaluate our orthogonal update mechanism against standard linear connections [2] by training models from scratch on several image classification benchmarks: CIFAR-10, CIFAR-100 [14] (both $32 \times 32$, 10/100 classes, 50k train/10k val. images), TinyImageNet [15] ($64 \times 64$, 200 classes, 100k train/10k val.), and ImageNet-1k [16] ($224 \times 224$, 1000 classes, $\approx$1.28M train/50k val.). Standard data augmentation strategies appropriate for each dataset and architecture were employed (full details in Appendix A.1).

**Architectures.** Our experiments utilize two primary architecture families: ResNetV2 [2] (employing standard configurations like -18, -34, -50, -101) and Vision Transformers (ViT) [5] (ViT-S: 384 hidden dim, 6 layers, 6 heads; ViT-B: 768 hidden dim, 12 layers, 12 heads). For ViTs, input images are processed into patch embeddings with a [CLS] token and 1D positional embeddings, and the [CLS] token's output representation is used for classification. Due to computational constraints, our ImageNet-1k evaluations focused on ViT models. All models use a final linear classification layer.

**Training Setup.** We follow standard protocols for each architecture family. ResNetV2 models were trained for 200 epochs with SGD [27] (batch size 128); ViTs for 300 epochs with AdamW [28] (batch sizes 1024 for smaller datasets and 4096 for ImageNet-1k). Full hyperparameters (e.g., learning-rate schedules and warmup, weight decay/momentum or AdamW betas, label smoothing, gradient clipping), augmentations, patch sizes, and compute details are provided in Appendix A.2. We use a stability constant $\epsilon = 10^{-6}$ by default; sensitivity and mixed-precision notes are deferred to Sec. 4.7 and Appendix G.

We emphasize that our ViT experiments follow the challenging academic paradigm of training *from scratch* on ImageNet-1k, rather than leveraging pre-training on large-scale proprietary datasets. This widely adopted setting requires strong regularization (e.g., MixUp, CutMix) to ensure robust convergence. Consequently, our results should be interpreted within the context of a rigorous paired comparison, designed to isolate the precise impact of the residual update mechanism.

### 4.2 Image Classification

We evaluated our orthogonal update mechanism by training various ResNetV2 and Vision Transformer (ViT) architectures from scratch on standard image classification datasets using consistent augmentation strategies per architecture family (see Appendix A for full settings). Due to computational constraints, our ImageNet-1k experiments focused on ViT models. All performance metrics are presented in Tab. 2.

Table 2: **Mean ± std. of Val Acc@1 (%) on 5 runs.** Results are averaged over the 5 best validation epochs from each run. Performance of standard Linear updates is compared against our orthogonal updates: Orthogonal-F (Feature-wise) and Orthogonal-G (Global). Due to computational constraints, ImageNet-1k experiments focused on ViTs.

| Architecture | Connection | Dataset (Acc@1 % mean ± std.) | | | |
|---|---|---|---|---|---|
| | | CIFAR-10 | CIFAR-100 | TinyImageNet | ImageNet-1k |
| ViT-S | Linear | $89.82_{\pm0.34}$ | $71.92_{\pm0.24}$ | $51.30_{\pm0.40}$ | $70.76_{\pm0.26}$ |
| | Orthogonal-F | $\mathbf{90.61}_{\pm0.21}$ | $\mathbf{73.86}_{\pm0.31}$ | $\mathbf{52.57}_{\pm0.71}$ | $\mathbf{72.53}_{\pm0.49}$ |
| ViT-B | Linear | $87.28_{\pm0.41}$ | $68.25_{\pm0.88}$ | $55.29_{\pm0.71}$ | $73.27_{\pm0.58}$ |
| | Orthogonal-F | $\mathbf{91.93}_{\pm0.08}$ | $\mathbf{75.07}_{\pm0.43}$ | $\mathbf{57.87}_{\pm0.37}$ | $\mathbf{77.05}_{\pm0.21}$ |
| ResNetV2-18 | Linear | $95.06_{\pm0.15}$ | $77.67_{\pm0.28}$ | $62.04_{\pm0.29}$ | |
| | Orthogonal-F | $\mathbf{95.26}_{\pm0.12}$ | $\mathbf{77.87}_{\pm0.27}$ | $\mathbf{62.65}_{\pm0.14}$ | — |
| | Orthogonal-G | $95.25_{\pm0.11}$ | $77.53_{\pm0.19}$ | $62.32_{\pm0.22}$ | |
| ResNetV2-34 | Linear | $95.49_{\pm0.09}$ | $78.92_{\pm0.31}$ | $64.61_{\pm0.24}$ | |
| | Orthogonal-F | $\mathbf{95.75}_{\pm0.13}$ | $\mathbf{78.97}_{\pm0.04}$ | $\mathbf{65.46}_{\pm0.30}$ | — |
| | Orthogonal-G | $95.53_{\pm0.12}$ | $78.71_{\pm0.24}$ | $65.38_{\pm0.35}$ | |
| ResNetV2-50 | Linear | $\mathbf{94.75}_{\pm0.09}$ | $\mathbf{77.90}_{\pm0.24}$ | $63.74_{\pm0.18}$ | |
| | Orthogonal-F | $94.71_{\pm0.11}$ | $77.43_{\pm0.10}$ | $\mathbf{64.22}_{\pm0.28}$ | — |
| | Orthogonal-G | $\mathbf{94.75}_{\pm0.10}$ | $77.56_{\pm0.34}$ | $64.40_{\pm0.36}$ | |
| ResNetV2-101 | Linear | $\mathbf{94.86}_{\pm0.05}$ | $77.72_{\pm0.33}$ | $63.77_{\pm0.52}$ | |
| | Orthogonal-F | $94.80_{\pm0.13}$ | $\mathbf{78.50}_{\pm0.26}$ | $65.78_{\pm0.22}$ | — |
| | Orthogonal-G | $94.75_{\pm0.13}$ | $78.37_{\pm0.19}$ | $\mathbf{65.87}_{\pm0.23}$ | |

Orthogonal updates demonstrated particularly strong benefits in ViT models. For instance, ViT-B with our method achieved a notable **+3.78 pp** increase in Acc@1 on ImageNet-1k over the baseline. In contrast, while generally positive, the performance gains from orthogonal updates on ResNetV2 architectures appeared more modest relative to those in ViTs. We posit that this difference in efficacy may, in part, be related to the "dimensionality-to-depth ratio", $\gamma$, characteristics of these architecture families. See Appendix F for the definition and estimates of $\gamma$.

Table 3: **ViT-S LR sweep on CIFAR-10/100**: Val Acc@1 (mean±std over 3 runs).

| | CIFAR-10 | | CIFAR-100 | |
|---|---|---|---|---|
| LR | Linear | Orthogonal | Linear | Orthogonal |
| $5 \times 10^{-4}$ | $90.41_{\pm0.15}$ | $\mathbf{90.56}_{\pm0.32}$ | $71.46_{\pm0.59}$ | $\mathbf{72.48}_{\pm0.37}$ |
| $8 \times 10^{-4}$ | $90.45_{\pm0.16}$ | $\mathbf{90.91}_{\pm0.33}$ | $71.13_{\pm0.92}$ | $\mathbf{72.99}_{\pm0.29}$ |
| $1 \times 10^{-3}$ | $89.95_{\pm0.36}$ | $\mathbf{90.36}_{\pm0.15}$ | $70.59_{\pm0.80}$ | $\mathbf{73.11}_{\pm0.39}$ |
| $2 \times 10^{-3}$ | $84.85_{\pm1.07}$ | $\mathbf{87.38}_{\pm0.42}$ | $62.17_{\pm1.21}$ | $\mathbf{69.71}_{\pm0.91}$ |
| $5 \times 10^{-3}$ | $66.09_{\pm1.29}$ | $\mathbf{72.20}_{\pm2.66}$ | $42.61_{\pm2.82}$ | $\mathbf{48.24}_{\pm2.50}$ |

**Learning-rate robustness.** We sweep the initial learning rate on a logarithmic grid ($5 \times 10^{-4}$ to $5 \times 10^{-3}$) while keeping all other hyperparameters identical to the main recipe (optimizer, schedule, regularization, and augmentation). As summarized in Tab. 3, the orthogonal update consistently outperforms the linear baseline.

### 4.3 Representational Metrics

Relative to the linear-residual baseline, the orthogonal update (i) increases *Effective Rank* and *Spectral Entropy* (broader, more uniform spectra), (ii) markedly reduces *feature standard deviation* (more stable activations), and (iii) yields a nontrivial linear CKA [29], suggesting structural differences. The metrics are defined as follows:

- **Effective Rank** is defined as $\exp(H)$, where $H := -\sum_i p_i \log p_i$ is the spectral entropy, and $p_i := \lambda_i / \sum_j \lambda_j$ are the normalized eigenvalues of the feature covariance matrix.

Table 4: **Representational metrics** on ViT–B (ImageNet–1k).

| Metric | Linear | Orthogonal | $\Delta$ |
|---|---|---|---|
| Effective Rank | 572.9 | 599.9 | **+4.7%** |
| Spectral Entropy | 6.512 | 6.539 | **+0.41%** |
| CKA (linear) | — | 0.546 | — |
| Feature Std. Dev. | 0.407 | 0.193 | **-52.5%** |

- **CKA (linear)** [29]: A similarity metric for two representations $X, Y$ defined as $\frac{\|X^\top Y\|_F^2}{\|X^\top X\|_F \|Y^\top Y\|_F}$.
- **Feature Std. Dev.**: The average per-feature standard deviation, $\frac{1}{d} \sum_{k=1}^{d} \text{Std}(F_{\cdot k})$.

Table 5: **Connection-type transfer on ViT-S (with reset).** Each run trains $300 \to 300$ epochs on the same dataset. Optimizer/LR scheduler are re-initialized at the switch. Optimizer states are *cleared* at switch points to isolate connection effects.

| Dataset | Start Arch.$\to$ End Arch. | | Acc@1 (%) | Acc@5 (%) |
|---|---|---|---|---|
| CIFAR-10 | Linear | $\to$ Linear | $92.78_{\pm0.06}$ | $99.74_{\pm0.03}$ |
| | Linear | $\to$ Orthogonal | $92.88_{\pm0.14}$ | $99.72_{\pm0.03}$ |
| | Orthogonal | $\to$ Linear | $93.89_{\pm0.12}$ | $\mathbf{99.75}_{\pm0.04}$ |
| | Orthogonal | $\to$ Orthogonal | $\mathbf{94.10}_{\pm0.12}$ | $\mathbf{99.73}_{\pm0.04}$ |
| CIFAR-100 | Linear | $\to$ Linear | $74.22_{\pm0.13}$ | $92.26_{\pm0.13}$ |
| | Linear | $\to$ Orthogonal | $74.02_{\pm0.24}$ | $91.96_{\pm0.17}$ |
| | Orthogonal | $\to$ Linear | $\mathbf{75.63}_{\pm0.17}$ | $\mathbf{92.91}_{\pm0.17}$ |
| | Orthogonal | $\to$ Orthogonal | $75.38_{\pm0.35}$ | $92.20_{\pm0.13}$ |
| TinyImageNet | Linear | $\to$ Linear | $53.24_{\pm0.13}$ | $75.25_{\pm0.21}$ |
| | Linear | $\to$ Orthogonal | $52.14_{\pm0.18}$ | $74.20_{\pm0.20}$ |
| | Orthogonal | $\to$ Linear | $\mathbf{54.58}_{\pm0.10}$ | $\mathbf{76.45}_{\pm0.24}$ |
| | Orthogonal | $\to$ Orthogonal | $53.88_{\pm0.29}$ | $75.34_{\pm0.23}$ |

Table 6: **Continuous training (without reset).** ViT-S trained for $150 \to 150$ epochs without mid-training re-initialization.

| Dataset | Start Arch.$\to$ End Arch. | | Acc@1 (%) | Acc@5 (%) |
|---|---|---|---|---|
| CIFAR-10 | Linear | $\to$ Linear | $89.82_{\pm0.34}$ | $99.65_{\pm0.03}$ |
| | Linear | $\to$ Orthogonal | $91.00_{\pm0.14}$ | $99.66_{\pm0.02}$ |
| | Orthogonal | $\to$ Linear | $\mathbf{93.18}_{\pm0.15}$ | $\mathbf{99.72}_{\pm0.03}$ |
| | Orthogonal | $\to$ Orthogonal | $90.61_{\pm0.21}$ | $99.69_{\pm0.03}$ |
| CIFAR-100 | Linear | $\to$ Linear | $71.92_{\pm0.24}$ | $92.11_{\pm0.18}$ |
| | Linear | $\to$ Orthogonal | $71.64_{\pm0.56}$ | $91.96_{\pm0.24}$ |
| | Orthogonal | $\to$ Linear | $\mathbf{74.14}_{\pm0.35}$ | $\mathbf{92.69}_{\pm0.19}$ |
| | Orthogonal | $\to$ Orthogonal | $73.86_{\pm0.31}$ | $92.23_{\pm0.26}$ |
| TinyImageNet | Linear | $\to$ Linear | $51.30_{\pm0.40}$ | $75.19_{\pm0.66}$ |
| | Linear | $\to$ Orthogonal | $50.78_{\pm0.42}$ | $73.91_{\pm0.32}$ |
| | Orthogonal | $\to$ Linear | $\mathbf{53.33}_{\pm0.62}$ | $\mathbf{76.06}_{\pm0.46}$ |
| | Orthogonal | $\to$ Orthogonal | $52.57_{\pm0.71}$ | $75.33_{\pm0.57}$ |

## 4.4 Ablation on Adapting Connection Types

We study how training adapts when switching residual–connection types in ViT-S on CIFAR-10/100 and TinyImageNet. We evaluate two regimes that disentangle potential confounds:

- **with optimizer/scheduler reset at the switch** to isolate the pure effect of changing the connection.
- **without re-initialization**, changing only the residual-connection type while preserving the optimizer state and learning-rate scheduler (continuous training).

### 4.4.1 Switching with optimizer/scheduler reset

As summarized in Tab. 5, **O→O** consistently surpasses **L→L**. Moreover, **O→L** often matches or exceeds **O→O**, whereas **L→O** shows no consistent gains over **L→L**. This pattern suggests that orthogonal updates are most beneficial early in training, while later linear updates can refine the learned representations.

### 4.4.2 Switching without optimizer re-initialization

Across datasets in Tab. 6, the same trend holds: **O→O** ≥ **L→L**, and **O→L** remains competitive or best despite preserving optimizer/scheduler states, while **L→O** does not reliably improve on **L→L**. Hence, the gains are attributable to the update geometry rather than optimizer resets.

**Joint interpretation.** The two regimes yield consistent conclusions. Orthogonal updates confer benefits early (cf. **O→O** ≥ **L→L**), and **O→L** is competitive or best even without re-initialization, indicating that the gains stem from the update geometry rather than optimizer resets. In contrast, **L→O** shows no systematic improvement over **L→L**.

### 4.5 Orthogonal Connection Probability

To investigate the impact of stochastic orthogonal connections, we conducted an ablation study on ViT-S using TinyImageNet (N=3 runs). In this setup, each residual connection employs an orthogonal update with a probability $\pi$, otherwise defaulting to the standard linear connection. We varied $\pi$ from 0.0 (all linear) to 1.0 (all orthogonal).

The effect of varying $\pi$ is visualized in Fig. 5. A clear and statistically positive trend is evident for both Acc@1 and Acc@5 accuracy (statistical p-value < 0.05, based on Pearson correlation shown in Fig. 5). This strongly suggests that consistent application of orthogonal updates (i.e., higher $\pi$) is beneficial for ViT-S performance on this task.

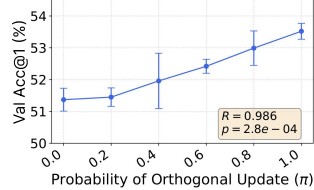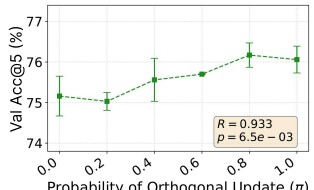

Figure 5: **Effect of orthogonal update probability** $\pi$ on ViT-S performance on TinyImageNet (N=3). Error bars represent $\pm 1$ standard deviation. The Pearson correlation coefficient ($R$) and its p-value between $\pi$ and accuracy are displayed in each subplot, indicating a positive correlation.

### 4.6 Orthogonal Layer Pattern Ablation

This ablation study investigates the impact of the placement and number of orthogonal connections within ViT-S model on the TinyImageNet dataset. We compare various patterns of applying the orthogonal update to specific layer indices (or groups thereof) against a baseline model ("None") with linear residual updates and a model where all layers employ orthogonal residual updates ("All (0-5)"). Tab. 7 shows that applying orthogonal connections to a greater number of layers tends to yield improved performance.

Table 7: **Performance on ViT-S layer patterns for orthogonal updates on TinyImageNet (N=3 runs).** Layer indices indicate where orthogonal connections were applied. "None" is the linear update. "All" applies orthogonal update to all 6 layers. Metrics are Acc@1 and Acc@5 (%) with mean ± std.

| Metric (%) | Applied Orthogonal Connection Layer Indices | | | | | | | |
|---|---|---|---|---|---|---|---|---|
| | None | 0,1 | 2,3 | 4,5 | 0,1,2,3 | 0,1,4,5 | 2,3,4,5 | All (0-5) |
| Acc@1 | 51.88±0.42 | 51.45±0.16 | 51.32±0.63 | 52.20±0.27 | 51.55±0.30 | 52.44±0.38 | 52.14±0.15 | **52.98**±0.45 |
| Acc@5 | 75.20±0.44 | 75.01±0.23 | 74.79±0.29 | 75.62±0.41 | 75.05±0.25 | 75.19±0.09 | 75.17±0.14 | **75.93**±0.58 |

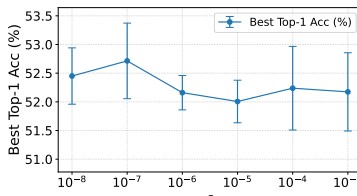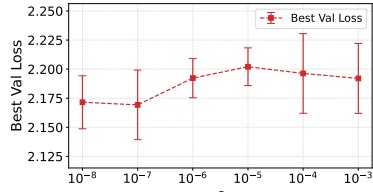

Figure 6: Effect of stability constant $\epsilon$ on ViT-S best validation performance. The x-axis represents $\epsilon$ on a logarithmic scale. Error bars indicate $\pm 1$ standard deviation across runs.

## 4.7 Stability Constant $\epsilon$

We examine the sensitivity of our method to the stability constant $\epsilon$, introduced in Eq. 3, which prevents division by zero during the orthogonal projection. We varied $\epsilon$ across several orders of magnitude, $\epsilon \in \{10^{-8}, 10^{-7}, 10^{-6}, 10^{-5}, 10^{-4}, 10^{-3}\}$, using ViT-S on TinyImageNet over five runs.

Fig. 6 visualizes the best Acc@1 and validation loss achieved for each $\epsilon$ value. We observed that the $\epsilon = 10^{-6}$ setting yielded results with the lowest standard deviation across the runs indicating greater stability and reproducibility. Prioritizing this stability, we adopted $\epsilon = 10^{-6}$ for the default constant throughout this paper, noting that its average performance remains competitive.

## 5 Conclusion

We revisited additive residual connections and showed that module outputs can contain a sizable component parallel to the residual stream. We introduced the *Orthogonal Residual Update*, which discards the parallel part and updates using the orthogonal component $f_\perp$. Across Vision Transformers and ResNetV2 on standard image-classification benchmarks, we observed improvements in generalization and distinct training dynamics with modest computational overhead. Given the ubiquity of residual connections, we hope this study encourages the community to probe residual-stream geometry at larger scales and in broader modalities.

**Limitations** Our experiments are limited by compute: models up to ViT-B and datasets up to ImageNet–1k, with non-exhaustive hyperparameter sweeps. We did not evaluate substantially deeper ResNetV2 variants on ImageNet–1k, web-scale datasets, or large language models, and our analysis focused on image classification. A deeper theoretical account of when/why orthogonal updates help, and their long-horizon stability, remains open.

**Future Work** Promising directions include (i) module-aware strategies (attention vs. MLP), (ii) schedules that mix early orthogonal updates with later linear ones, (iii) scaling studies to test the role of the width-to-depth ratio $\gamma$, and (iv) applications beyond classification (e.g., diffusion models and sequence modeling). We anticipate that community-scale evaluations will be particularly valuable given how widely residual connections are deployed.

## Acknowledgement

This work was partly supported by an Institute of Information & communications Technology Planning & Evaluation (IITP) grant funded by the Korean Government (MSIT) (No. RS-2020-II201361, Artificial Intelligence Graduate School Program (Yonsei University), No. RS-2024-00353131 and No. RS-2021-II211343, Artificial Intelligence Graduate School Program (Seoul National University)), the National Research Foundation of Korea(NRF) grant funded by the Korea government(MSIT)(RS-2024-00354218) and KAIT GPU project.

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

# A  Hyperparameters and Hardware

This section details the training details and data augmentation settings used for the Vision Transformer (ViT-S/-B) and ResNetV2 experiments presented in the main paper. Specific configurations for each model across the different datasets are provided in Tab. 8 (ViT-S/-B) and Tab. 9 (ResNetV2).

## A.1  Architecture Specific Hyperparameters

Table 8: Full training hyper-parameters of ViT-S/-B and data augmentations (mean/std. results in Tab. 2 are based on five independent runs with these settings).

| Hyperparameter | CIFAR-10 | CIFAR-100 | Tiny-ImageNet | ImageNet-1k |
|---|---|---|---|---|
| Epochs | 300 | 300 | 300 | 300 |
| Batch size | 1024 | 1024 | 1024 | 4096 |
| Base LR | $1\times10^{-3}$ | $1\times10^{-3}$ | $5\times10^{-4}$ | $5\times10^{-4}$ |
| Min LR | 0 | 0 | $5\times10^{-5}$ | $5\times10^{-5}$ |
| Optimizer | AdamW, $(\beta_1, \beta_2) = (0.9, 0.999)$, weight-decay $1 \times 10^{-4}$ | | | |
| LR scheduler | Cosine, 10-epoch linear warm-up | | | |
| Resolution (px) | 32 | 32 | 64 | 224 |
| Random crop | pad 4 | pad 4 | scale (0.8, 1.0) ratio (0.75, 1.33) | scale (0.08, 1.0) ratio (0.75, 1.33) |
| Patch size | 4×4 | 4×4 | 8×8 | 16×16 |
| Color jitter | brightness/contrast/saturation = 0.4, hue = 0.1 | | | |
| Horizontal flip | p = 0.5 | | | |
| MixUp $\alpha$ / prob | $\alpha = 0.8$, prob = 1.0 | | | |
| CutMix $\alpha$ | $\alpha = 1.0$ | | | |
| Label smoothing | $\varepsilon = 0.1$ | | | |
| Random erase | p = 0.25, scale (0.02, 0.33), ratio (0.3, 3.3) | | | |
| RandAug (N, M) | (9, 5) | (9, 5) | (9, 5) | (9, 9) |
| Normalization | dataset-specific mean/std | | | |

Table 9: Full training hyper-parameters and data augmentations for ResNetV2 models (e.g., ResNetV2-18, -34, -50, -101) on CIFAR and Tiny-ImageNet datasets. Mean/std. results in Tab. 2 are based on five independent runs with these settings.

| Hyperparameter | CIFAR-10 | CIFAR-100 | Tiny-ImageNet |
|---|---|---|---|
| Epochs | 200 | | |
| Batch size | 128 | | |
| Base LR | $1 \times 10^{-1}$ | | |
| Optimizer | SGD, momentum 0.9 | | |
| Weight decay | $5 \times 10^{-4}$ | | |
| LR scheduler | MultiStep, decay by 0.2 at epochs 80, 120 | | |
| Warm-up epochs | 0 (Not used) | | |
| Resolution (px) | 32 | 32 | 64 |
| Random crop | Pad 4px, random 32×32 crop | | Random 64×64 crop, scale (0.08-1.0), ratio (0.75-1.33) |
| Horizontal flip | p = 0.5 | | |
| Color jitter | — (Not used) | | |
| MixUp $\alpha$ / prob | — (Not used) | | |
| CutMix $\alpha$ | — (Not used) | | |
| Label smoothing | — (Not used) | | |
| Random erase | — (Not used) | | |
| RandAug (N, M) | — (Not used) | | |
| Gradient Clipping | — (Not used) | | |
| Normalization | Dataset-specific mean/std | | |

## A.2 Computational Resources and Training Times

The experiments were conducted using a variety of NVIDIA GPUs. The following provides an overview of the typical hardware configurations and approximate training times for key models and datasets. Actual training times could vary based on the specific GPU models available and system load during experimentation.

- **ImageNet-1k Training (Vision Transformers):**
    - ViT-B: Trained on a system with 8x NVIDIA L40S GPUs for approximately 42 hours.
    - ViT-S: Trained on a system with 8x NVIDIA L40S GPUs for approximately 15 hours.

- **CIFAR-10/100 and Tiny ImageNet Training (Vision Transformers):**
    - ViT-B:
        * CIFAR-10/100: Typically trained on 2x NVIDIA H100 GPUs. Approximate training time was 1 hours per run.
        * Tiny ImageNet: Typically trained on 2x NVIDIA H100 GPUs. Approximate training time was roughly double that of the CIFAR experiments, around 2 hours per run.
    - ViT-S:
        * CIFAR-10/100: Typically trained on 8x NVIDIA RTX 3090 GPUs for approximately 1 hour per run.
        * Tiny ImageNet: Typically trained on 8x NVIDIA RTX 3090 GPUs for approximately 2 hours per run.

- **CIFAR-10/100 and Tiny ImageNet Training (ResNetV2 Models):**
    - All ResNetV2 models (ResNetV2-18, -34, -50, -101) were trained on systems equipped with 4x NVIDIA RTX 3090 GPUs.
    - As a representative example, a full training run for ResNetV2-34 on CIFAR-10 took approximately 18 hours.
    - Training times for other ResNetV2 model variants on Tiny ImageNet, and for all ResNetV2 variants on the CIFAR-10/100 datasets, scaled accordingly with model depth and dataset size.

All training times are approximate and reflect the end-to-end duration for the specified number of epochs.

# B Formal Derivation and Theoretical Guarantees

## B.1 Geometric Interpretation as an Exponential Map Approximation

Our update rule can be rigorously interpreted from the perspective of differential geometry. We consider the learned feature space as a high-dimensional, curved manifold $\mathcal{M}$. The orthogonal decomposition $f(\sigma(x_n)) = f_\parallel + f_\perp$ occurs in the tangent space $T_{x_n}\mathcal{M}$ at the point $x_n$ on this manifold, which is a vector space where Euclidean geometry locally applies.

The geometrically formal tool for moving from a point $x_n$ along a direction specified by a tangent vector $v \in T_{x_n}\mathcal{M}$ is the exponential map, $\exp_{x_n}(v) : T_{x_n}\mathcal{M} \to \mathcal{M}$. Our proposed update rule, $x_{n+1} = x_n + f_\perp(x_n)$, can be understood as a first-order Taylor approximation of this map:

$$x_{n+1} \approx \exp_{x_n}(f_\perp(x_n))$$

This linear approximation is highly accurate when the tangent vector $v = f_\perp(x_n)$ is small relative to the local curvature of the manifold. Our empirical results provide strong support for this condition. As shown in our paper's internal dynamics analysis (e.g., Figs. 10 and 11), the norm of our update vector, $\|f_\perp(x_n)\|$, consistently remains orders of magnitude smaller than the norm of the feature stream itself, $\|x_n\|$. This ensures the update constitutes a small, local step on the manifold where the Euclidean approximation is valid. Therefore, our method is not a naive application of Euclidean geometry but a computationally efficient mechanism that leverages the local structure of the learned manifold to guide feature exploration.

## B.2 Neural ODE Perspective: Norm Dynamics under Orthogonal Flow

We can also interpret our recursion as an explicit Euler discretization of a continuous-time (Neural ODE [30]) flow. Let $x(t)$ evolve according to

$$\dot{x}(t) = f_\perp(x(t)), \qquad f_\perp(x) = f(\sigma(x)) - s(x)\,x, \quad s(x) = \frac{\langle f(\sigma(x)), x\rangle}{\|x\|^2 + \epsilon}.$$

Then the squared norm satisfies the exact identity

$$\frac{d}{dt}\|x(t)\|^2 = 2\langle x(t), \dot{x}(t)\rangle = 2\langle x(t), f(\sigma(x(t))) - s(x(t))x(t)\rangle$$

$$= 2\langle x(t), f(\sigma(x(t)))\rangle \left(1 - \frac{\|x(t)\|^2}{\|x(t)\|^2 + \epsilon}\right) = 2\langle x(t), f(\sigma(x(t)))\rangle \frac{\epsilon}{\|x(t)\|^2 + \epsilon}. \quad (6)$$

In the idealized case $\epsilon = 0$ (exact orthogonal projection), Eq. (6) reduces to $\frac{d}{dt}\|x(t)\|^2 = 0$, i.e., the flow remains on the sphere $\|x(t)\| = \mathrm{const}$. For $\epsilon > 0$, the radial component is suppressed by the factor $\epsilon/(\|x(t)\|^2 + \epsilon)$, which is small whenever $\|x(t)\|^2 \gg \epsilon$.

Moreover, the discrete update $x_{n+1} = x_n + f_\perp(x_n)$ yields

$$\|x_{n+1}\|^2 = \|x_n\|^2 + 2\langle x_n, f_\perp(x_n)\rangle + \|f_\perp(x_n)\|^2,$$

so the first-order norm drift term vanishes when $\epsilon = 0$ (since $\langle x_n, f_\perp(x_n)\rangle = 0$), and for $\epsilon > 0$ it remains attenuated according to Eq. (6). Empirically, we observe stable feature norms and $\|f_\perp(x_n)\| \ll \|x_n\|$ throughout depth (see Figs. 10 and 11).

## B.3 Identity gradient path preservation

Let $f_\perp(x_n) = f(\sigma(x_n)) - s_n x_n$ with $s_n = \frac{\langle f(\sigma(x_n)), x_n\rangle}{\|x_n\|^2 + \epsilon}$ and update $x_{n+1} = x_n + f_\perp(x_n)$. Then

$$\frac{\partial x_{n+1}}{\partial x_n} = (1 - s_n)I - x_n(\nabla_{x_n} s_n)^\top + \frac{\partial f(\sigma(x_n))}{\partial x_n}, \quad (7)$$

$$= I + \left[\frac{\partial f(\sigma(x_n))}{\partial x_n} - \frac{\partial(s_n x_n)}{\partial x_n}\right] = I + \frac{\partial f_\perp(x_n)}{\partial x_n}. \quad (8)$$

Since the Jacobian equals the identity plus a residual term, the crucial identity path $I$ enabling unhindered gradient flow across depth is preserved, in the same sense as standard residual networks. The stability constant $\varepsilon > 0$ guarantees differentiability even when $\|x_n\|$ is small.

## B.4 Analysis of the Derivative Term $\frac{\partial(s_n x_n)}{\partial x_n}$

$$\frac{\partial(s_n x_n)}{\partial x_n} = \underbrace{s_n I}_{\substack{\text{Isotropic scaling;}\\\text{affects all components,}\\\text{incl. parallel to } x_n}} + \underbrace{\frac{x_n f^\top}{b}}_{\substack{\text{Output along } x_n;\\\text{magnitude from } f^\top v;\\\text{propagates } f_\parallel \text{ and } f_\perp \text{ effects}}}$$

$$+ \underbrace{\frac{x_n (J_f^\top x_n)^\top}{b}}_{\substack{\text{Output along } x_n;\\\text{via Jacobian } J_f \text{ interaction}}} - \underbrace{\frac{2\langle x_n, f\rangle}{b^2} x_n x_n^\top}_{\substack{\text{Rank-1 update along } x_n;\\\text{modulates } x_n\text{-parallel strength}}} , \tag{9}$$

where $b = \|x_n\|_2^2 + \epsilon$.

Deconstructing these terms reveals how information related to the $x_n$-parallel component of $f(\sigma(x_n))$ (denoted $f_\parallel$) and its $x_n$-orthogonal component ($f_\perp^{\text{true}}$, to distinguish from the update $f_\perp(x_n)$) can propagate through the gradients:

- The first term, $s_n I$, isotropically scales all components of an input perturbation, including any component parallel to $x_n$. Since $s_n = \langle x_n, f\rangle/b$, its magnitude is directly influenced by the alignment of $f$ with $x_n$.

- The second term, $\frac{x_n f^\top}{b}$, always maps an input perturbation to an output in the direction of $x_n$. If we decompose $f = f_\parallel + f_\perp^{\text{true}}$, this term becomes $\frac{x_n(f_\parallel + f_\perp^{\text{true}})^\top}{b} = \frac{\langle x_n, f\rangle/\|x_n\|^2}{b} x_n x_n^\top + \frac{1}{b} x_n (f_\perp^{\text{true}})^\top$. The first part explicitly carries the $f_\parallel$ influence via a rank-1 update in the $x_n$ direction. The second part shows how even the true orthogonal component $f_\perp^{\text{true}}$ contributes to a gradient term that effectively "leaks" into the $x_n$ direction.

- The third term, $\frac{x_n(J_f^\top x_n)^\top}{b}$, similarly produces outputs only along $x_n$, with its magnitude depending on complex interactions involving the module's Jacobian $J_f$. This term can mix influences from both parallel and orthogonal components of $f$ as captured by $J_f$.

- The fourth term, $-\frac{2\langle x_n, f\rangle}{b^2} x_n x_n^\top$, is a rank-1 update that directly adjusts components in the $x_n$ direction, scaled by the alignment $\langle x_n, f\rangle$. It often acts as a form of negative feedback or dampening for the $x_n$-parallel contributions.

Thus, the derivative $\frac{\partial f_\perp(x_n)}{\partial x_n} = \frac{\partial f(\sigma(x_n))}{\partial x_n} - \frac{\partial(s_n x_n)}{\partial x_n}$ does not imply that the $x_n$-parallel aspects of $f(\sigma(x_n))$ are ignored during backpropagation. Instead, their influence is intricately incorporated into the learning dynamics. The presence of $\epsilon$ in $b$ also means that $s_n$ (and consequently $f_\perp(x_n)$ itself, as shown in Eq. (10)) does not perfectly nullify the parallel component, allowing for controlled modulation of information along the $x_n$ stream, which can be beneficial for norm stability (further discussed in Sec. B.5).

$$\langle x_n, f_\perp(x_n)\rangle = \langle x_n, f(\sigma(x_n))\rangle \frac{\epsilon}{\|x_n\|^2 + \epsilon} \tag{10}$$

## B.5 Proof of the Non-Vanishing of the Parallel Component

Suppose that $\epsilon \approx 0$ but, not exactly zero.

$$\frac{\partial(s_n x_n)}{\partial x_n} \cdot f_\parallel = \left(s_n I + \frac{\langle x_n, f\rangle}{b} + \frac{\langle x_n, J_f^\top x_n\rangle}{b} - \frac{2\langle x_n, f\rangle}{b^2}\|x\|_2^2\right) \cdot (s_n x_n) \tag{11}$$

$$= 2s_n^2 x_n + \frac{\langle x_n, J_f^\top x_n\rangle}{b} s_n x_n - 2s_n^2 x_n \tag{12}$$

$$= \frac{\langle J_f x_n, x_n\rangle}{b} s_n x_n \tag{13}$$

The expression is zero if $\begin{cases} s_n = 0 & \text{(i.e., } f(\sigma(x_n)) \perp x_n) \\ \langle J_f x_n, x_n\rangle = 0 & \text{(Jacobian action is orthogonal)} \\ x_n = 0 & \text{(zero input)} \end{cases}$

1. **Condition 1:**

   Experimentally this almost never occurs. To have $s_n = 0$ requires

   $$\langle f\big(\sigma(x_n)\big),\, x_n \rangle = 0,$$

   i.e. the output vector of $f$ at $\sigma(x_n)$ must be exactly orthogonal to the basis direction $x_n$. In practice—due to the continuous nature of $f$, model noise, and floating-point effects—there is virtually always some nonzero component along any chosen direction, so the inner product is never exactly zero.

2. **Condition 2:**

   This orthogonality condition holds precisely when the Jacobian $J_f$ of the mapping $f$ is orthogonal to the basis direction $x_n$. In practice, whenever perturbations along $x_n$ do not influence the differential behaviour of $f$, the gradient and $x_n$ are orthogonal, and the condition is met.

3. **Condition 3**

   This is the trivial "zero input" case, which simply never shows up in real data. Requiring the entire input vector $x_n$ to be identically zero before the nonlinearity $\sigma$ is a degenerate scenario—and one that your network or dataset will practically never produce—so you won't observe this condition in experiments.

# C  Ablation Studies

## C.1  Learning Curves of ResNetV2 and Ours-G Differing the Initialization Method

We evaluated the robustness of our *Orthogonal Residual Update* to various initialization methods. Specifically, we implemented three initialization approaches: Xavier [31], Kaiming He [32], and Orthogonal [19] initialization for the network weights. For all experiments, the biases of the convolution layers and batch normalization layers were initialized to zero, while the batch normalization weights were set to one. Previous research has established that initialization methods significantly impact model performance and convergence properties [33–35]. Therefore, we conducted a comparative analysis of the variation of the top-5 validation accuracy (reported in Fig. 7) and final accuracy values (reported in Tab. 10) between the original identity connection and our proposed approach. Our experimental configuration followed the hyperparameter settings from ResNetV2 [2], with training conducted on the CIFAR-100 dataset for 60,000 steps.

Table 10: Mean ± std oftop-1 (Acc@1) and top-5 (Acc@5) accuracy from 5 independent runs.We used ResNetV2-50-G for OURS.

| Initialization | Connection | Acc@1(%) | Acc@5(%) |
|---|---|---|---|
| Orthogonal[19] | ResNetV2 | $77.40_{\pm 0.33}$ | $93.00_{\pm 0.20}$ |
| | Ours-G | $77.22_{\pm 0.24}$ | $93.28_{\pm 0.15}$ |
| Xavier[31] | ResNetV2 | $77.43_{\pm 0.37}$ | $93.07_{\pm 0.26}$ |
| | Ours-G | $77.09_{\pm 0.31}$ | $93.07_{\pm 0.09}$ |
| Kaiming He[32] | ResNetV2 | $77.78_{\pm 0.16}$ | $92.82_{\pm 0.20}$ |
| | Ours-G | $77.54_{\pm 0.20}$ | $93.37_{\pm 0.12}$ |

Table 11: Hyperparameter settings of the experiment varying the initialization method. We used identical hyperparameter setting across all experiments.

| Hyperparameter | Values |
|---|---|
| Epochs | 200 |
| Batch size | 128 |
| Base LR | $1 \times 10^{-1}$ |
| Optimizer | SGD, momentum 0.9 |
| Weight decay | $5 \times 10^{-4}$ |
| LR scheduler | MultiStep, decay by 0.1 at epochs 80, 120 |
| Warm-up epochs | 0 (Not used) |
| Resolution (px) | 32 |
| Random crop | Pad 4px, random $32 \times 32$ crop |
| Horizontal flip | p = 0.5 |
| Color jitter | — (Not used) |
| MixUp $\alpha$ / prob | — (Not used) |
| CutMix $\alpha$ | — (Not used) |
| Label smoothing | — (Not used) |
| Random erase | — (Not used) |
| RandAug (N, M) | — (Not used) |
| Gradient Clipping | 1.0 |
| Normalization | Dataset-specific mean/std |

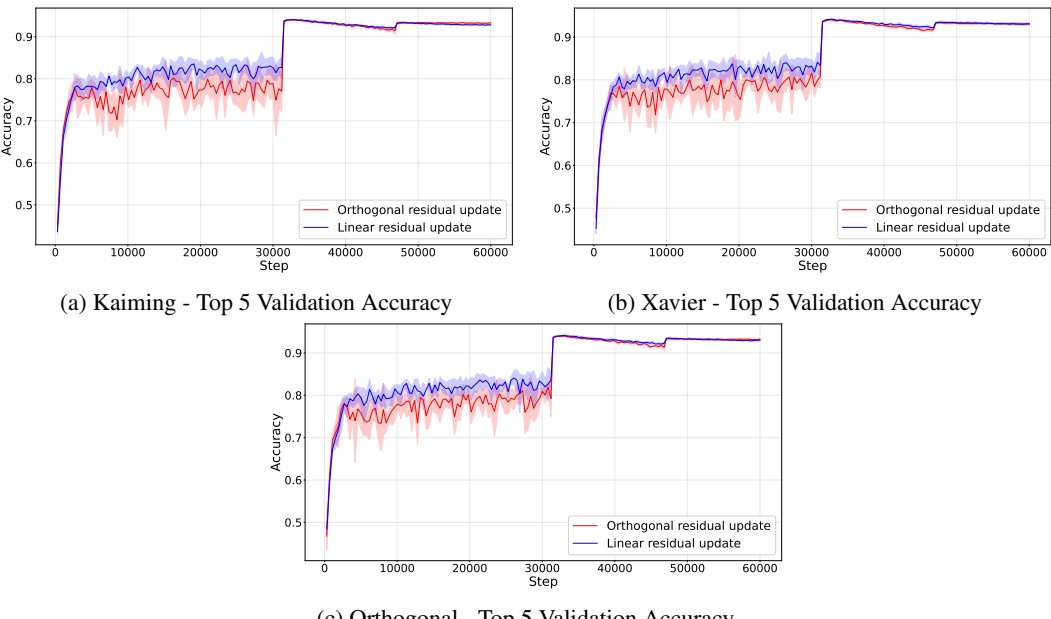

(a) Kaiming - Top 5 Validation Accuracy

(b) Xavier - Top 5 Validation Accuracy

(c) Orthogonal - Top 5 Validation Accuracy

Figure 7: Mean value of validation accuracy over training steps from 5 independent runs for different initialization methods: (a) Kaiming Top-5, (b) Xavier Top-5, (c) Orthogonal Top-5. The shaded regions show the intervals one standard deviation above and below the mean.

## C.2 $f$ Norm Exploding in ResNetV2 with Orthogonal Updates

We observed a notable phenomenon in ResNetV2 architectures during later training stages: the L2 norm of the output of the final convolutional layer module, $f(x_n)$, tended to increase dramatically, particularly with linear residual updates. Interestingly, this explosion in $\|f(x_n)\|^2$ was not always accompanied by a similarly large input stream norm $\|x_n\|^2$. While this behavior was noted across various ResNetV2 configurations, we focus here on ResNetV2-34 trained on Tiny ImageNet, using the same experimental settings as detailed in Tab. 9.

Fig. 8 illustrates this trend, comparing the evolution of the module output norm ($\|f(x_n)\|^2$), its component parallel to the input stream ($\|f_\parallel(x_n)\|^2$), and its component orthogonal to the input stream ($\|f_\perp(x_n)\|^2$) for both linear and orthogonal residual updates in the final block. For linear updates, a significant increase, especially in $\|f(x_n)\|^2$ and often its parallel component $\|f_\parallel(x_n)\|^2$, is evident at later steps.

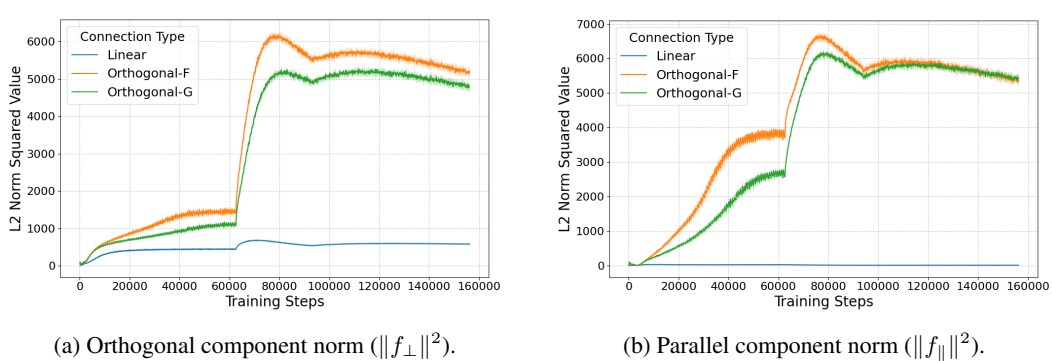

(a) Orthogonal component norm ($\|f_\perp\|^2$).

(b) Parallel component norm ($\|f_\parallel\|^2$).

Figure 8: Component norms of the final convolutional module's output in ResNetV2-34 on Tiny ImageNet **without** a final LayerNorm. Both plots compare Linear, Orthogonal-F (feature-wise), and Orthogonal-G (global) updates. (X-axis: Training Steps, Y-axis: L2 Norm Squared). The explosion or significant growth of component norms is visible, particularly for the parallel component under certain updates.

This phenomenon is suspected to be related to the accumulation of the parallel component $f_\parallel(x_n)$ over layers, as discussed in Sec. 3.4, which can influence the overall module output $f(x_n)$ when the module itself is not explicitly regularized for orthogonality. To mitigate this, we implemented a simple solution: adding a LayerNorm [36] (LN) layer immediately before the final classifier head, applied to the output of the last residual stream (i.e., after the final global average pooling in ResNetV2). This LayerNorm effectively normalizes the activations passed to the classifier.

As shown in Fig. 9, the introduction of this final LayerNorm significantly stabilized the norms of $f(x_n)$ and its components for the final convolutional module, especially for the linear residual updates, preventing the previously observed explosion.

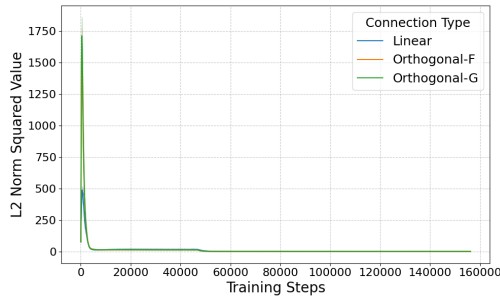
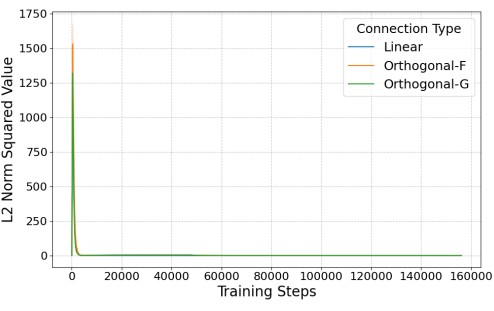

(a) Orthogonal component norm ($\|f_\perp\|^2$).      (b) Parallel component norm ($\|f_\parallel\|^2$).

Figure 9: Component norms of the final convolutional module's output in ResNetV2-34 on Tiny ImageNet **with** a final LayerNorm applied before the classifier head. (X-axis: Training Steps, Y-axis: L2 Norm Squared). The norm explosion is mitigated, and component norms remain more stable throughout training.

With this LayerNorm fix applied to ResNetV2-34 on Tiny ImageNet, the model with linear residual updates achieved a top-1 accuracy of 65.88%±0.22%, while the model with our Orthogonal Residual Updates (Feature-wise) achieved 65.83%±0.38%. Tab. 12 summarizes these indicative results. While the norm explosion was primarily an issue for linear updates, applying LayerNorm is a common practice and ensures fair comparison.

Table 12: Indicative top-1 accuracy (%) of ResNetV2-34 on TinyImageNet with and without the final LayerNorm (LN) before the classifier. Values are mean ± std. from 5 runs. † denote metrics from Tab. 2

| Configuration | Connection | Top-1 Acc. (%) |
|---|---|---|
| W/O final LN† | Linear | 64.61±0.24 |
| | Orthogonal-F | 65.46±0.30 |
| | Orthogonal-G | 65.38±0.35 |
| W/ final LN | Linear | 65.88±0.22 |
| | Orthogonal-F | 65.83±0.38 |
| | Orthogonal-G | 65.66±0.15 |

This investigation suggests that while our orthogonal updates inherently promote more stable norm dynamics within the residual blocks (as discussed in Sec. 3.4), careful consideration of normalization at the network's extremity, particularly before the classifier, can be beneficial for all types of residual connections in deep CNNs to prevent potential instabilities arising from the final layers.

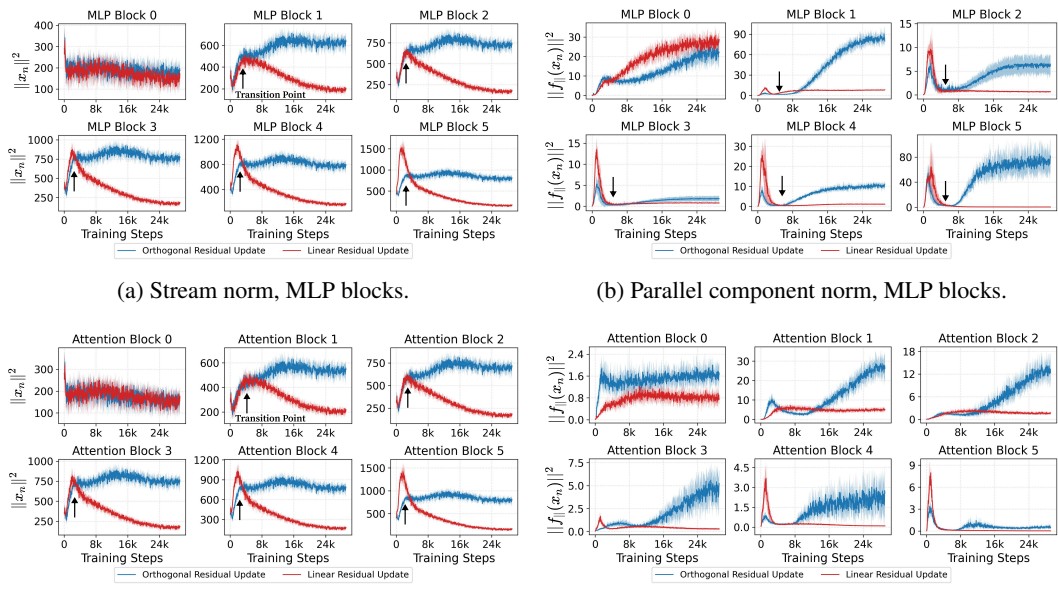

(a) Stream norm, MLP blocks.

(b) Parallel component norm, MLP blocks.

(c) Stream norm, Attention blocks.

(d) Parallel component norm, Attention blocks.

Figure 10: **Internal dynamics (ViT-S, TinyImageNet, 5 seeds).** Each subfigure shows blocks 0–5 (MLP top, Attention bottom). **Ours** denotes orthogonal updates; **Linear** denotes the standard residual. **(a,c)** After the *Transition Point*, orthogonal updates stabilize the stream norm $\|x_n\|$, whereas linear updates typically exhibit a post-transition decrease. **(b,d)** The parallel component energy $\|f_\|(\sigma(x_n))\|^2$ follows distinct layer-wise profiles for linear vs. orthogonal updates.

## D   Additional Details on Internal Norm Dynamics

This appendix consolidates the block–level internal dynamics of ViT–S (Tiny ImageNet, 5 seeds). We present two complementary views: (i) stream and alignment signals in Fig. 10 (stream norm $\|x_n\|^2$ and cosine $\cos(x_n, f(\sigma(x_n)))$), and (ii) component energies in Fig. 11, where we plot the squared L2 norm of the orthogonal component $\|f_\perp(\sigma(x_n))\|^2$ with the cosine panels shown alongside for context. Placed together, these figures provide a consolidated, self–contained view of the dynamics discussed in Sec. 3.4.

**General observations.**   Across layers, the orthogonal update (blue) maintains a substantial orthogonal–component energy $\|f_\perp(\sigma(x_n))\|^2$ throughout training (Fig. 11b, 11d), whereas the linear residual (red) exhibits a progressive reduction, most prominently in deeper blocks. This trend aligns with the post–Transition–Point behavior in Fig. 10: the linear pathway shows a peak–and–decay of the stream norm and a divergence in cosine, while the orthogonal pathway stabilizes the stream norm and avoids collapse of $\|f_\perp\|^2$. Early–layer nuances exist (e.g., MLP block 0 under linear may hold $\|f_\perp\|^2$ longer before declining), but the overall pattern is consistent: orthogonal updates preserve novel (orthogonal) directions, whereas linear updates tend to diminish them over training.

**Parallel component dynamics.**   While explicit panels for $\|f_\|(\sigma(x_n))\|^2$ are omitted, its evolution can be inferred from the joint trends of $\|f_\perp(\sigma(x_n))\|^2$ (Fig. 11) and cosine (Fig. 10). Under *orthogonal* updates, $\|f_\perp\|^2$ remains substantial and the cosine increases in later training, which is consistent with the full module output gradually accumulating a parallel part even though the update step $f_\perp$ is strictly orthogonal. This follows from the decomposition $f(\sigma(x_n)) = s_n x_n + f_\perp(\sigma(x_n))$: gradients through the $s_n x_n$ term allow the parallel component to grow, whereas the update path discards it. By contrast, the *linear* pathway shows a sustained reduction of $\|f_\perp\|^2$ together with a peak–then–decay pattern in $\|x_n\|^2$ (Fig. 10), indicating a shift of capacity toward stream-aligned directions. A Jacobian-based derivation is provided in Appendix B.4.

**The Transition Point and cosine dynamics.**   The *Transition Point* in Fig. 10 (the inflection of $\|x_n\|^2$ with the onset of divergence in $\cos(x_n, f(\sigma(x_n)))$) coincides with a redistribution of component

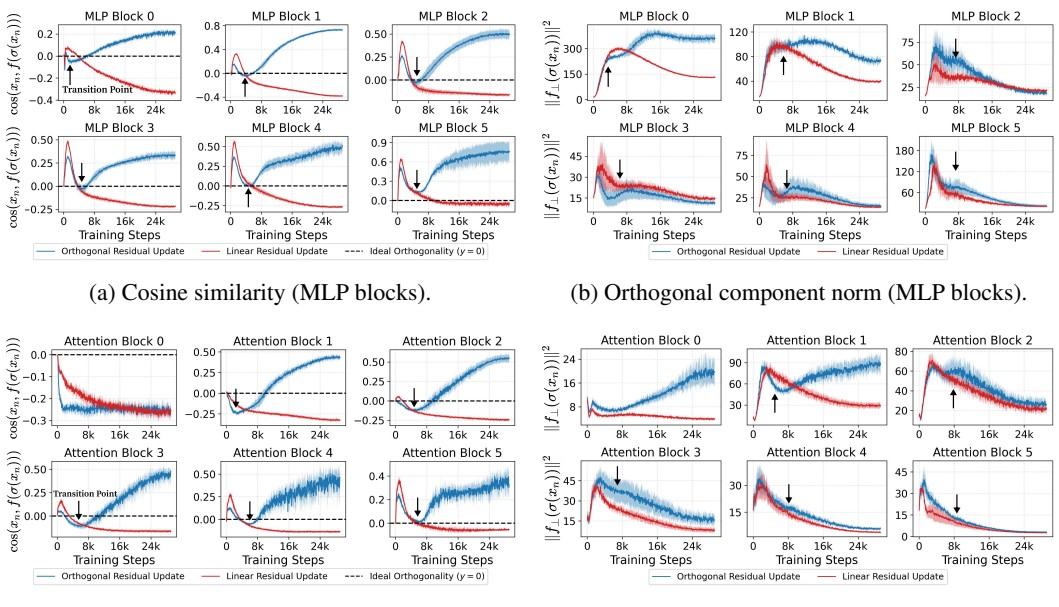

(a) Cosine similarity (MLP blocks).

(b) Orthogonal component norm (MLP blocks).

(c) Cosine similarity (Attention blocks).

(d) Orthogonal component norm (Attention blocks).

Figure 11: **Cosine similarity and orthogonal norm across layers (ViT–S, Tiny ImageNet, 5 seeds).** Each subfigure aggregates blocks 0–5 for MLP (top) and Attention (bottom). Ours (orthogonal updates) vs. Linear (standard residual). Orthogonal updates preserve the orthogonal component energy, while cosine trajectories diverge around the Transition Point. For stream norms and a broader view of alignment, see Fig. 10.

energies in Fig. 11. After this point, the **linear** pathway typically exhibits a peak–then–decay of $\|x_n\|^2$ together with a sustained reduction of $\|f_\perp(\sigma(x_n))\|^2$, most clearly in deeper MLP/Attention blocks, indicating a drift toward stream–aligned updates and diminished novelty. In contrast, the **orthogonal** pathway stabilizes $\|x_n\|^2$ while *preserving* $\|f_\perp\|^2$ across depth; the subsequent rise in $\cos(x_n, f(\sigma(x_n)))$ is explained by the full module output gradually *accumulating* a parallel portion $s_n x_n$ even though the update step remains strictly orthogonal. Empirically, earlier blocks (e.g., 0–1) often maintain or mildly increase both $\|f_\perp\|^2$ and alignment, producing a gentle cosine uptick, whereas later blocks (e.g., 3–5) show a dominant parallel share with non–negligible $\|f_\perp\|^2$, sustaining directional diversity without collapse. This reconciles the seemingly counterintuitive growth of cosine under orthogonal updates with the preserved orthogonal energy, consistent with the mechanism in Sec. 3.4.

**Implications for layer-wise application of orthogonality.** The depth–dependent trends of $\|f_\|(\sigma(x_n))\|^2$ and $\|f_\perp(\sigma(x_n))\|^2$ suggest that the effect of orthogonality is not uniform across layers. If one considers *targeted* application (e.g., early–only or late–only; see Sec. 4.6), these component dynamics provide a guideline: early blocks often sustain larger $\|f_\perp\|^2$ and can seed diversified features that propagate forward, whereas later blocks may benefit from preserving non–negligible orthogonal energy while alignment grows. That said, our main configuration *applies orthogonality throughout* the network (Tab. 2), and the observed persistence of $\|f_\perp\|^2$ across many layers, together with the ablation results in Tab. 7, indicates that full–depth application is a robust default; selective patterns can recover a substantial portion of the gains but generally do not surpass the all–layers setting.

# E  Extended Comparisons: Unified Residual Family with Fixed Start Conditions

We consider the unified residual family

$$x_{n+1} = x_n + \rho_\ell\big(\sin\theta_\ell\, f_\|(x_n) + \cos\theta_\ell\, f_\perp(x_n)\big), \qquad \rho_\ell \geq 0,\ \theta_\ell \in \left[-\tfrac{\pi}{2}, \tfrac{\pi}{2}\right], \tag{14}$$

and track, for each block $\ell$, the phase–plane coordinates $(\rho_\ell\sin\theta_\ell,\ \rho_\ell\cos\theta_\ell)$, which correspond to the *parallel* and *orthogonal* shares, respectively. Two canonical start conditions are used:

$$(\rho_\ell, \theta_\ell) = (\sqrt{2}, \tfrac{\pi}{4}) \quad \text{(Linear start; gives } \rho\sin\theta = \rho\cos\theta = 1\text{)},$$
$$(\rho_\ell, \theta_\ell) = (1, 0) \qquad \text{(Orthogonal start; gives } \rho\sin\theta = 0,\ \rho\cos\theta = 1\text{)}.$$

We then train under Eq. (14) and visualize trajectories over steps.

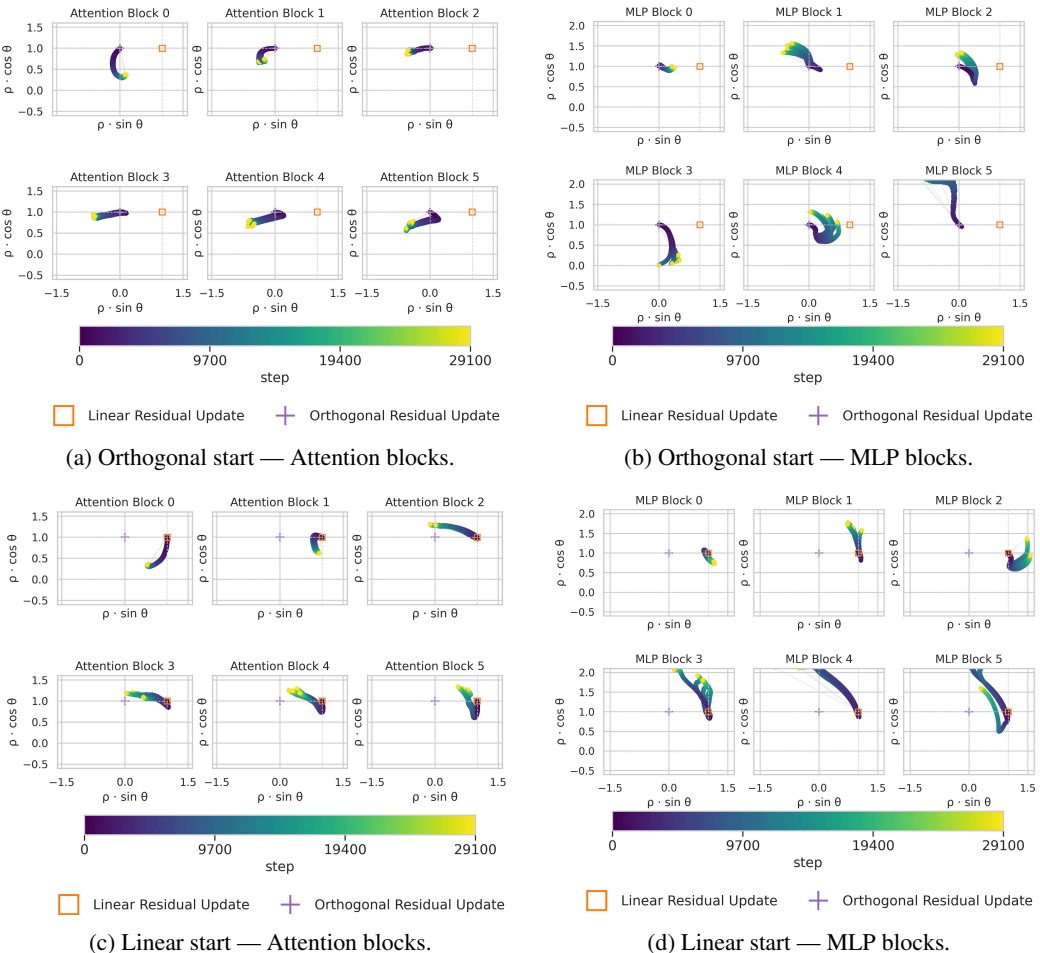

(a) Orthogonal start — Attention blocks.

(b) Orthogonal start — MLP blocks.

(c) Linear start — Attention blocks.

(d) Linear start — MLP blocks.

Figure 12: **Unified-residual phase plane with fixed start conditions.** Axes are $\rho\sin\theta$ (parallel share, $x$) and $\rho\cos\theta$ (orthogonal share, $y$); color encodes training steps. Crosshairs indicate the canonical start points: $(0, 1)$ for the orthogonal start and $(1, 1)$ for the linear start. Trajectories reveal how each block balances parallel/orthogonal contributions over training.

**Observations.** For clarity, the $x$–axis encodes the *parallel share* $\rho\sin\theta$ and the $y$–axis the *orthogonal share* $\rho\cos\theta$. Orthogonal–start runs begin at $(0, 1)$ on the $y$–axis and retain a sizable orthogonal share across depth; linear–start runs begin at $(1, 1)$ and typically drift toward lower $y$ (reduced orthogonal share), especially in deeper blocks. Attention and MLP trace different paths, but both show the same trend: when updates remove the parallel component, the orthogonal share does not collapse, consistent with the internal–dynamics analysis in Sec. 3.4.

# F   $\gamma$: Architectural Ratio of Width to Depth

**Definition of $\gamma$.**   For Vision Transformers and other Transformer-based architectures (e.g., GPT-2 [37], LLaMA [6], T5 [38], DiT [39]), we define

$$\gamma_{\text{Transformer}} = \frac{d_{\text{model}}}{L_{\text{blocks}}}.$$

For ResNets, we define

$$\gamma_{\text{ResNet}} = \frac{D_{\text{avg}}}{B_{\text{total}}}, \quad D_{\text{avg}} = \frac{\sum_{k \in \text{stages}} B_k C_k^{\text{out}}}{B_{\text{total}}},$$

where $B_k$ is the number of residual blocks in stage $k$, $C_k^{\text{out}}$ is that stage's output-channel dimensionality (i.e., the dimension of the feature map to which the residual $f(x_n)$ is added), and $B_{\text{total}} = \sum_k B_k$. Equivalently,

$$\gamma_{\text{ResNet}} = \frac{\sum_{k \in \text{stages}} B_k C_k^{\text{out}}}{B_{\text{total}}^2}.$$

**On the ratio $\gamma$.**   Before analyzing inter-family differences, we first consider the ratio $\gamma$, which heuristically characterizes representational width per sequential processing block; a formal definition is provided immediately below. For models of comparable size, ViT-S (22.0M) yields $\gamma \approx 384/6 \approx 64$, whereas ResNetV2-34 (21.8M) yields $\gamma \approx 3776/16^2 \approx 14.8$. The lower $\gamma$ of ResNetV2-34 suggests a more compact per-block representational space, potentially leaving less directional redundancy for standard updates to exploit and partly explaining the more modest gains observed.

**Comparative $\gamma$ Values.**   Tab. 13 lists the architectural parameters and calculated $\gamma$ values for models discussed in this paper and several external reference models.

Table 13: Comparison of $\gamma$ values across various model architectures. $d_{model}$ denotes hidden dimension; $L$ denotes number of layers/blocks. For ResNets, $B_k$ is blocks per stage, $C_k^{out}$ is output channels per stage, $B_{total}$ is total residual blocks, and $D_{avg} = (\sum B_k C_k^{out})/B_{total}$. The $\gamma$ for ResNets is $D_{avg}/B_{total}$.

| Family | Model | $d_{model}$ or $D_{avg}$ | $L$ or $B_{total}$ | $\gamma$ | Notes |
|---|---|---|---|---|---|
| *Models from this paper's experiments* | | | | | |
| ResNetV2 | ResNet-18 | 240.00 ($D_{avg}$) | 8 | 30.00 | Basic blocks |
| | ResNet-34 | 236.00 ($D_{avg}$) | 16 | 14.75 | Basic blocks |
| | ResNet-50 | 944.00 ($D_{avg}$) | 16 | 59.00 | Bottleneck blocks |
| | ResNet-101 | 985.21 ($D_{avg}$) | 33 | 29.85 | Bottleneck blocks |
| ViT | ViT-S (ours) | 384 ($d_{model}$) | 6 | 64.00 | Used in paper |
| | ViT-B (ours) | 768 ($d_{model}$) | 12 | 64.00 | Used in paper |
| *External Reference Models* | | | | | |
| ResNet | ResNet-1001 | 37.33 ($D_{avg}$) | 168 | 0.22 | [1] |
| ViT | ViT-S (std.) | 384 ($d_{model}$) | 12 | 32.00 | [5] |
| | ViT-L | 1024 ($d_{model}$) | 24 | 42.67 | |
| | ViT-H | 1280 ($d_{model}$) | 32 | 40.00 | |
| Language | GPT-2 XL | 1600 ($d_{model}$) | 48 | 33.33 | 1.5B params [37] |
| Models | LLaMA-7B | 4096 ($d_{model}$) | 32 | 128.00 | [6] |
| | T5-XL | 2048 ($d_{model}$) | 24 | 85.33 | Encoder/Decoder [38] |
| Diffusion | DiT-XL/2 | 1152 ($d_{model}$) | 28 | 41.14 | [39] |

**Discussion.**   The $\gamma$ values presented in Tab. 13 span a wide range, reflecting diverse architectural philosophies. As noted in the main paper, the ViT models used in our experiments (ViT-S $\gamma = 64$, ViT-B $\gamma = 64$) have substantially higher $\gamma$ values compared to ResNetV2-34 ($\gamma = 14.75$) and are comparable to or higher than ResNetV2-18 ($\gamma = 30.00$) and ResNetV2-101 ($\gamma \approx 29.85$). Interestingly, ResNetV2-50 ($\gamma = 59.00$) also exhibits a high $\gamma$ value, approaching that of our ViT

models. This is primarily due to its bottleneck architecture: our $\gamma$ definition for ResNets calculates $D_{avg}$ based on the *expanded output channel dimensions* of the blocks (i.e., the dimensionality of the stream $x_n$ where the residual $f(x_n)$ is added) and $B_{total}$ as the count of these residual blocks. Since bottleneck blocks in ResNetV2-50 feature significantly wider output dimensions (resulting in $D_{avg} = 944$) than the basic blocks in models like ResNetV2-34 ($D_{avg} = 236$) for the same number of residual blocks ($B_{total} = 16$), its calculated $\gamma$ is consequently higher.

It is important to acknowledge a nuance in this $\gamma$ definition when applied to bottleneck ResNets versus Transformers or basic-block ResNets. While our $\gamma$ calculation consistently uses the dimensionality of the stream where the residual sum occurs, a single bottleneck block internally performs multiple distinct convolutional operations with varying channel dimensions (e.g., a 1x1 channel reduction, a 3x3 convolution in the narrower space, and a 1x1 channel expansion). If one were to define an "effective $\gamma$" that accounts for this internal sequential processing or averages across these varying internal channel widths (rather than primarily considering the wide output where the addition happens), the $\gamma$ value for bottleneck architectures like ResNetV2-50 could be interpreted as being substantially lower. Thus, while our current $\gamma$ definition highlights the width of the feature space available for the residual addition, it may not fully capture the operational "compactness" imposed by the narrower processing stages within each bottleneck block, especially when comparing against Transformer layers that typically maintain a more uniform processing width throughout.

Despite this definitional consideration, the consistently calculated high $\gamma$ for ResNetV2-50 indicates a large dimensional space for its residual updates. The precise interplay between this $\gamma$ metric, specific architectural inductive biases (e.g., local receptive fields in CNNs vs. global attention in Transformers), and the observed efficacy of the *Orthogonal Residual Update* warrants further nuanced investigation. The diverse $\gamma$ values seen in other reference models—such as the extremely low $\gamma$ for the deep and narrow ResNet-1001 (CIFAR config) versus the very high $\gamma$ values for large language models like LLaMA-7B and T5-XL—further emphasize that $\gamma$ is one of several factors influencing how different residual update mechanisms might perform across various architectural paradigms.

# G   PyTorch Implementation

This section provides example PyTorch implementations for the channel-wise and global orthogonalization functions used to compute the orthogonal component $f_\perp(x_n)$ from a module output $f(x_n)$ and an input stream $x_n$. These functions encapsulate the core logic of Eq. 3 from the main paper.

```python
def _orthogonal_channel(x: torch.Tensor, f_x: torch.Tensor, dim: int, eps:
    torch.Tensor) -> torch.Tensor:
    """
    Orthogonal residual connection (channel-wise).
    x   : residual stream tensor
    f_x : module output tensor (e.g., from Attention, MLP, or Conv if channel-wise)
    dim : dimension along which to compute orthogonality (e.g., channel dimension)
    eps : small epsilon tensor for numerical stability
    """
    # Ensure eps is on the same device as x if it's a tensor
    eps = eps.to(x.device)

    dot_product = (x * f_x).sum(dim, keepdim=True)
    norm_x_squared = (x * x).sum(dim, keepdim=True).float() + eps

    # Ensure scale is cast back to original dtype if x was float16/bfloat16
    scale_factor = (dot_product / norm_x_squared).to(dtype=x.dtype)

    projection_onto_x = scale_factor * x
    f_orthogonal = f_x - projection_onto_x

    return f_orthogonal
```

Listing 1: PyTorch function for channel-wise orthogonalization.

```python
def _orthogonal_global(x: torch.Tensor, f_x: torch.Tensor, dim: int, eps:
    torch.Tensor) -> torch.Tensor:
    """
    Orthogonal residual connection (global).
    x   : residual stream tensor
    f_x : module output tensor (e.g., from a convolutional block)
    dim : starting dimension for flattening (all subsequent dims will be flattened)
    eps : small epsilon tensor for numerical stability
    """
    original_shape = x.shape
    # Convert negative dim to positive for consistent unsqueezing later
    positive_dim_idx = dim if dim >= 0 else len(original_shape) + dim

    eps = eps.to(x.device)

    x_flattened = x.flatten(start_dim=positive_dim_idx) # [B, C, H, W] -> [B,
        C*H*W] if dim=1 (for NCHW)
    f_x_flattened = f_x.flatten(start_dim=positive_dim_idx) # or [B, D] if already
        2D from start_dim

    # Sum over the flattened dimensions (which is now dim=1 if start_dim was 1, or
        the last dim)
    # For clarity, explicitly use dim=-1 for sum over the last (flattened) dimension
    dot_product = (x_flattened * f_x_flattened).sum(dim=-1, keepdim=True)
    norm_x_squared = (x_flattened * x_flattened).sum(dim=-1, keepdim=True).float()
        + eps

    scale_factor = (dot_product / norm_x_squared).to(dtype=x.dtype)

    # Reshape scale_factor to allow broadcasting with original x shape
```

```
    # It needs to have trailing dimensions of size 1 to match x's rank
        post-flattening start_dim
    num_dims_to_unsqueeze = len(original_shape) - (positive_dim_idx + 1) # +1
        because dot_product keeps one dim
    for _ in range(num_dims_to_unsqueeze):
        scale_factor = scale_factor.unsqueeze(-1)

    projection_onto_x = scale_factor * x # Broadcasting happens here
    f_orthogonal = f_x - projection_onto_x

    return f_orthogonal
```

Listing 2: PyTorch function for global orthogonalization.

The `dim` argument in `_orthogonal_channel` typically refers to the channel dimension (e.g., `dim=1` for NCHW tensors) where orthogonality is computed independently for each spatial location or token. For `_orthogonal_global`, `dim` specifies the starting dimension from which the tensor is flattened before computing a single global projection scale per batch element; for instance, for an NCHW tensor, `dim=1` would flatten C, H, and W dimensions together. The choice between these depends on the layer type and desired granularity of orthogonalization as discussed in Sec. 3.3 of the main paper.

