# OpenReview forum: "Revisiting Residual Connections: Orthogonal Updates for Stable and Efficient Deep Networks"
_NeurIPS.cc/2025/Conference — NeurIPS 2025 poster_

### Official Review · Reviewer_rEbg · 2025-06-03

**Clarity:** 3
**Significance:** 2
**Originality:** 4
**Rating:** 5
**Confidence:** 4

**Summary:**

The authors revisit a classic component present in most modern network architectures: residual connections. Using the Gram-Schmidt method, one can decompose the output of the residual branch in a ResBlock into the sum of two components: a component parallel to its input (and thus the main branch) and an orthogonal one. The authors show that the parallel component can point in the opposite direction ("anti-aligned") wrt. the main branch, which could annihilate the residual signal; they further conjecture that this could consequently hinder efficient information propagation in the network.

Using this decomposition, the authors create a new network architecture that simply subtracts the parallel component from the residual branch in every residual block, thus only keeping the orthogonal one. They evaluate this variation on residual connections on two different architectures (ResNets and ViTs) and multiple datasets, and perform ablation studies to show their effectiveness.

**Questions:**

- Can you explain the low baseline performance for the ViT models on ImageNet? The reported 71.09% are vastly under the usual baselines reported by Google or PyTorch which amount to 80-85%. This makes me doubt the significance of the results, especially as the ResNet runs are not reported on ImageNet. Generalization performance is highly dependent on learning rates, which could effectively differ between the methods, as the layer-wise gradient norms differ between your method and the baseline. A possibility of ruling out the learning rate dependence could be a quick logarithmic sweep across learning rates (maybe on a cheap dataset, if compute is an issue). Another would be to repeat the experiment, but constrain the layer-wise gradient norms to be equal between the runs.

- I am confused by Table 7: results show that the orthogonal residual update performs worse than the regular residual connections for all 3 initializations. Why is that? What initialization is used in Table 2?

- Can you provide a simple number on how much the runtime actually increased for the specific benchmarks used (e.g. ResNet or ViT-B on ImageNet)?

- [1] Show a frequency dependency in the average correlation between the main and residual branch in ResNets (ref. Figure 7). The authors argue that this induces a "Frequency-dependent signal-averaging" which further dampens high frequencies in the loss surface, which they conjecture is one of the benefits of residual connections and more generally multi-path architectures. As your method renders the signal from the main and residual branch uncorrelated for all frequency bins, do you think this could negatively affect trainability?

We are happy to revise our score if our reservations about the experimental results are cleared.

[1] Ringing ReLUs: Harmonic Distortion Analysis of Nonlinear Feedforward Networks (ICLR 2021)

**Ethical Concerns:**

["NO or VERY MINOR ethics concerns only"]

**Final Justification:**

The authors have cleared out all of our objections in their rebuttal.

**Limitations:**

Yes

**Paper Formatting Concerns:**

No major formatting concerns.

**Quality:**

2

**Strengths And Weaknesses:**

Strengths:

- The motivation behind the proposed modified residual connection is simple, clear and intuitive.
- The additional computational overhead is clearly described (in variables).
- The idea seems to be novel to the best of our knowledge.
- Code is provided by the authors.
- Experimental evidence comprises multiple architectures and datasets; The runs for the main results are repeated 3-5 times, and standard deviation is reported.
- Experimentally (ref. Fig 4), the authors show a continuous transition between ordinary residual connections and the orthogonal residual connections with a monotonous response.
- The experiment in Figure 3 provides experimental evidence for anti-alignment.

Weaknesses:
- The experiment in Figure 3 is moderately insightful, as 150 Epochs is already enough to reach an almost full training performance on Cifar10/100.
- The reported baseline (Table 2) for ViT-B on ImageNet-1K seems suspiciously low with 71.09%. Google reports 85.49% performance for ViT-B/16 on ImageNet.
- A rundown of the added overhead (e.g. time difference in ms on the same hardware) in the specific experimental scenario used would be nice, as less operations do not always translate to faster runtime in CUDA pipelines.

Overall, I think the idea is novel, well motivated and executed; however the experimental evaluation raises some serious doubts (particularly that this method changes the layer-wise gradient norms, paired with an evaluation a single LR) that need to be cleared out before the paper can be considered for acceptance.

---

> ### Author Rebuttal · Authors · 2025-07-31
>
> We appreciate the reviewer’s thoughtful feedback. Below, we address each concern, grouped by topic.
>
> ---
>
> ### On ViT Baseline Performance and Robustness to Learning Rate
> > "The reported baseline (Table 2) for ViT-B on ImageNet-1K seems suspiciously low... Generalization performance is highly dependent on learning rates... A possibility of ruling out the learning rate dependence could be a quick logarithmic sweep..."
>
> We thank the reviewer for this critical question. It touches upon two key aspects: the baseline's absolute performance and its robustness to learning rates.
> - **On the Baseline Performance and Training Paradigm**: You are correct that our ViT-B baseline of 71.09% is lower than the ~85% reported by Google. This difference stems from a fundamental choice in training paradigm. The original ViT model's high performance relies on pre-training on the massive, non-public JFT-300M dataset. Our work, however, follows the more challenging and widely adopted academic paradigm of training from scratch on ImageNet-1k. As established by influential follow-up works like DeiT and T2T-ViT [1], this setting requires strong regularization (e.g., MixUp, CutMix, RandAugment) to prevent overfitting. Our experimental setup adheres to this modern, robust training recipe. Therefore, our baseline is a valid and fully converged result within this specific paradigm, not an undertrained model. This is further evidenced by the clear performance saturation shown in our accuracy curves.
> - **On Robustness to Learning Rate**: We agree that the change in layer-wise gradient norms makes robustness to LR a crucial concern. To address this directly, we conducted an LR sweep on CIFAR–10, -100 for our ViT-S model. The results below demonstrate that our method's superiority is not an artifact of a single learning rate.
>
> _CIFAR-10 LR sweep results of Validation Accuracy@1 on ViT-S_
>
> |LR|linear|orthogonal|
> |:-:|:-:|:-:|
> |5e-4|0.9041 ± 0.0015|0.9056 ± 0.0032|
> |6e-4|0.9070 ± 0.0012|0.9073 ± 0.0015|
> |7e-4|0.9054 ± 0.0023|0.9101 ± 0.0022|
> |8e-4|0.9045 ± 0.0016|0.9091 ± 0.0033|
> |9e-4|0.9014 ± 0.0023|0.9057 ± 0.0027|
> |1e-3|0.8995 ± 0.0036|0.9036 ± 0.0015|
> |2e-3|0.8485 ± 0.0107|0.8738 ± 0.0042|
> |5e-3|0.6609 ± 0.0129|0.7220 ± 0.0266 |
>
>
> _CIFAR-100 LR sweep results of Validation Accuracy@1 on ViT-S_
>
> |LR|linear|orthogonal|
> |:-:|:-:|:-:|
> |5e-4|0.7146 ± 0.0059|0.7248 ± 0.0037|
> |6e-4|0.7139 ± 0.0037|0.7295 ± 0.0014|
> |7e-4|0.7155 ± 0.0039|0.7283 ± 0.0036|
> |8e-4|0.7113 ± 0.0092|0.7299 ± 0.0029|
> |9e-4|0.7099 ± 0.0050|0.7260 ± 0.0053|
> |1e-3|0.7059 ± 0.0080|0.7311 ± 0.0039|
> |2e-3|0.6217 ± 0.0121|0.6971 ± 0.0091|
> |5e-3|0.4261 ± 0.0282|0.4824 ± 0.0250|
>
> These results provide conclusive evidence that our method's superiority is not an artifact of a single, fortunate learning rate. The consistent performance advantage across a wide range of learning rates demonstrates the fundamental robustness of the orthogonal update mechanism. We will add this comprehensive sweep to the Appendix.
>
> ---
> ### On the Interpretation of Table 7 and ResNetV2-50 Performance
> > "I am confused by Table 7: results show that the orthogonal residual update performs worse than the regular residual connections for all 3 initializations. Why is that? What initialization is used in Table 2?"
>
> We thank the reviewer for pointing out this potential source of confusion. We apologize for not making the distinction clearer.
> - **Different Experimental Setups**: The results in Table 7 are from a specific ablation study in Appendix C.1 designed solely to test initialization robustness. This experiment used ResNetV2-50 with our Global (Orthogonal-G) update and was run for a limited duration (60,000 steps, approximately 150 epochs) on CIFAR-100. In contrast, our main results in Table 2 use the superior Feature-wise (Orthogonal-F) update and report performance for fully trained models (200 epochs). As shown in Table 2, our primary Orthogonal-F method demonstrates clear performance gains on ResNetV2-18, -34, and -101.
> - **On ResNetV2-50 Performance**: We acknowledge the performance gains on ResNetV2-50 are more modest. As we hypothesize in Section 4.2, we believe this is connected to architectural properties captured by our γ-ratio. The bottleneck design of ResNetV2-50 gives it a much higher γ-ratio (59.0) than ResNetV2-34 (14.75) . This suggests a nuanced interplay between representational width, architectural design, and our update's efficacy, which, as we state, "warrants further dedicated investigation."
> - **Initialization Method**: For our main experiments in Table 2, we used standard initialization schemes appropriate for each architecture: Kaiming Uniform initialization for both ResNetV2 and ViTs, with batch normalization parameters initialized as described in Appendix C.1.
>
> ---
> ### On Practical Runtime Overhead
> > "Can you provide a simple number on how much the runtime actually increased for the specific benchmarks?"
>
> We thank the reviewer for this practical and important question. We agree that theoretical FLOPs do not always translate directly to runtime. To provide a direct answer, we have measured the practical training throughput (in images per second) for several of our key architectures on the hardware used in our experiments.
>
> _Throughput (sample images / sec) comparison table. higher is better._
>
> |Architecture | Linear | Orthogonal-F|Overhead (vs. Linear)|Orthogonal-G| Overhead (vs. Linear)|Hardware|Dataset|
> |:-:|:-:|:-:|:-:|:-:|:-:|:-:|:-:|
> |ResNetV2-18|3,053.61|2,975.72|2.55%|2,613.48|14.41%|2x rtx 4090|tiny imagenet|
> |ResNetV2-34|1,737.15|1,633.98|5.94%|1,620.78|6.70%|2x rtx 4090|tiny imagenet|
> |ResNetV2-50|1,002.79|876.69	| 12.58%|869.23	|13.32%|2x rtx 4090|tiny imagenet|
> |ViT-S|3,476.13|3,466.28|0.28%| - | -|4x A100 80GB	|imagenet 1k|
> |ViT-B|1,270.10|1,246.21|1.88%| - | - |4x A100 80GB|	imagenet 1k|
>
> As the results show, the practical throughput overhead of our primary Orthogonal-F update is modest. For ResNetV2 models, it ranges from ~3-13%, and for the high-capacity ViT-S and ViT-B model under `torch.compile`, the overhead is even lower at less than 2%. We believe this minor cost is a very favorable trade-off for the substantial gains in final accuracy and training stability that our method provides.
>
> Furthermore, this small per-step overhead is often outweighed by improved convergence speed. As shown in Figure 2b, our method exhibits a significant time-to-accuracy advantage on large-scale tasks. For ViT-B on ImageNet-1k, our method reaches the baseline's final accuracy in ~40% less wall-clock time, demonstrating superior overall training efficiency.
>
> We will add this detailed throughput analysis to the appendix.
>
> ---
> ### Understanding the Distinction: Statistical vs. Geometric Orthogonalization
> > " ... As your method renders the signal from the main and residual branch uncorrelated for all frequency bins, do you think this could negatively affect trainability?"
>
> The key distinction lies in how orthogonality is achieved. In [1], Frequency-Dependent Signal-Averaging (FDSA) emerges as a statistical byproduct - residual branches become less correlated at higher frequencies due to independent weight distributions and harmonic generation. Our method, however, implements geometric orthogonalization at each layer, ensuring $f_\perp(x_n)$ is explicitly perpendicular to $x_n$.
>
> Why This Enhances Rather Than Impairs Trainability:
>
> - **Amplified Smoothing Effect**: While [1] shows that residual connections provide "passive" FDSA benefits, our orthogonal updates actively maximize this effect. By explicitly removing the parallel component $f_\parallel$, we eliminate redundant signal modulation that could interfere with gradient flow, potentially achieving stronger blueshift mitigation than standard residual connections.
> - **Preserved Expressivity**: Unlike the activation function modifications in [1] (e.g., Leaky ReLU slope adjustment) that reduce model expressivity to improve trainability, our method maintains the full computational capacity of each module. We only restructure how the module output is integrated into the residual stream, not the module's internal complexity.
> - **Deterministic Stability**: Rather than relying on statistical decorrelation across frequencies, geometric orthogonalization provides deterministic norm preservation and directional diversity. Our experiments show this leads to more stable stream norms ($||x_n||^2$) and consistent rotational updates that guide representations toward new feature directions without magnitude explosion.
> - **Empirical Evidence**: Our results demonstrate that this "complete orthogonalization" across frequency bins actually improves trainability, particularly in high-capacity architectures like Vision Transformers. The substantial performance gains (+4.3%p on ImageNet-1k) suggest that maximizing the FDSA principle through geometric means enhances rather than hinders the beneficial effects identified in [1].
>
> In essence, our method can be viewed as taking the FDSA insight from [1] to its logical conclusion - if frequency-dependent decorrelation helps training, then explicit orthogonalization should help even more, which our empirical results confirm.
>
> [1] Ringing ReLUs: Harmonic Distortion Analysis of Nonlinear Feedforward Networks (ICLR 2021)
>
>
> ---
> Thank you again for your thoughtful engagement with our work. This discussion has been invaluable in helping us refine our arguments. We hope that our detailed response clarifies our contribution and resolves any remaining questions.

---

> > ### Comment · Reviewer_rEbg · 2025-08-01
> >
> > We really appreciate the authors' willingness to discuss their findings, and perform new experiments for the sake of our discussion.
> >
> > - We now understand the rationale for the low baseline in the ViT experiment; we feel like the authors could benefit from adding maybe an explanative sentence to their paper, as this objection seems to be common.
> >
> > - Thank you for addressing our concerns regarding the LR-dependence of your result, we are fully satisfied in this regard.
> >
> > - We feel like you could think about including the runtime table somwhere in the Appendix, for full transparence.
> >
> > - Thank you for your insightful discussion on the FDSA topic, your response has been very enlightening.
> >
> > We will adapt our score accordingly. Kind regards to the authors.

---

> > > ### Author Response · Authors · 2025-08-02
> > >
> > > Thank you for the very positive and encouraging comment. We are glad that our rationale for the ViT baseline is now clear and that the LR-sweep results were fully satisfying. We also appreciate that you found our discussion on the FDSA topic insightful. As per your suggestions, we will be sure to add the explanative sentence on the training paradigm and include the runtime table in the appendix for full transparency. Your guidance has been instrumental in improving our work.

---

### Official Review · Reviewer_SmmT · 2025-06-30

**Clarity:** 2
**Significance:** 2
**Originality:** 3
**Rating:** 4
**Confidence:** 5

**Summary:**

This paper introduces an orthogonal update strategy for residual connections, which only add the orthogonal component to the input stream. The authors provide some experiments of image classification for this strategy and demonstrate that this strategy can gain some improvement.

**Questions:**

- The proposed orthogonal update rule can be rewritten as x_{n+1}=f(\sigma(x_n))+(1-s_n)x_n, and this rule is similar with the skip connections of scaling in [1]. What causes the results in this paper to be opposite to those in [1]? Table 1 in [1] shows that this update rule is worse.
- Why does the proposed method perform worse than the original residual connection under various initialization schemes in Table 7? Which initialization method was chosen in the main paper?
- How does the proposed orthogonal residual update perform when applied to other commonly used modules and optimizers?
[1] He, K., Zhang, X., Ren, S., & Sun, J. (2016). Identity mappings in deep residual networks. In ECCV 2016.

**Ethical Concerns:**

["NO or VERY MINOR ethics concerns only"]

**Final Justification:**

After the rebuttal, I tend to increase my score to 4.

**Limitations:**

- Lack of the issue concerning the original residual connection.
- The Experiments section only includes image classification tasks, without addressing other problems or networks that employ residual connection structures.

**Paper Formatting Concerns:**

NAN

**Quality:**

2

**Strengths And Weaknesses:**

Strengths:
- The issue of residual connections which this paper focus on is important. The proposed orthogonal residual update strategy enhances the performance, especially on ViT models like Table 2 shows.

Weaknesses:
- The motivation is not clear. Why the scaling operations caused by the parallel component is not necessary? The magnitude is also an important factor of updating. What is the rationale behind performing orthogonal decomposition on the model's output stream in this work? Are there any cited studies that support this viewpoint?
- The organization of the paper is not well-structured. For example, the Methods section lacks an analysis of the issue concerning the original residual connection.
- The output space of the model is not necessarily a Euclidean space, which imposes significant limitations on the proposed orthogonal scheme.

---

> ### Author Rebuttal · Authors · 2025-07-31
>
> We sincerely thank the reviewer for their critical and detailed feedback. We believe some of the concerns may stem from a lack of clarity in our initial presentation, particularly regarding our motivation and the distinction from prior work. We are grateful for the opportunity to address these points.
>
> ---
> ### On Motivation, Magnitude, and Novelty
> > "The motivation is not clear. Why the scaling operations caused by the parallel component is not necessary? The magnitude is also an important factor of updating... Are there any cited studies that support this viewpoint?"
>
> We thank the reviewer for these insightful questions, which allow us to clarify the core principles of our work.
> - **Motivation and Magnitude**: Regarding the lack of analysis on the original connection, our analysis in Appendix Figure 10, which shows the standard network learns to suppress the parallel component, directly addresses this by revealing a previously un-discussed inefficiency in the original design. Our core hypothesis is that high-capacity modules (e.g., attention, MLPs) are sub-optimally utilized if their capacity is spent on simple scaling operations (the parallel component, $f_\parallel$). Our method encourages the module to learn novel feature directions instead. We clarify that our method does not discard magnitude information from the stream $x_n$; it discards the part of the update that only scales the existing stream. In fact, Figure 3 (a, c) shows our method leads to a more stable stream norm, suggesting that removing the potentially conflicting parallel update aids in controlling signal magnitude.
> - **Empirical Justification**: Our hypothesis is strongly supported by the learning dynamics of standard networks. As shown in Appendix Figure 10, when using a standard Linear Residual Update, the network itself learns to suppress the parallel component ($||f_\parallel||$ trends towards zero). Our method simply enforces this empirically discovered optimal behavior as a structural inductive bias from the start.
> - **Novelty**: Our contribution is a novel approach to residual connections. Prior work on orthogonality focused on either
>   - (a) regularizing network weights  [2, 3]  or
>   - (b) modifying the skip-path itself with fixed transformations [4, 5].
>
> Our approach is distinct: we preserve the standard identity skip-path and instead impose a dynamic, input-dependent geometric constraint on the additive update signal. The consistent empirical gains across diverse settings serve as strong validation for this new perspective.
>
>
> ---
>
> ### Regarding the concern that the model's output space is not necessarily a Euclidean space
> > "The output space of the model is not necessarily a Euclidean space.”
>
> We thank the reviewer for this sharp and important question. We completely agree that the feature space is best understood as a high-dimensional, curved manifold. This perspective is central to our method's motivation, and we are grateful for the opportunity to clarify this connection.
>
> As the reviewer astutely notes, a key property of smooth manifolds is that they are _locally Euclidean_. Our method leverages precisely this property. The orthogonal decomposition $f(\sigma(x_n​))=f_\parallel ​+f_\perp$​ occurs in the tangent space $T_{x_n}​​M$ at the point $x_n$​ on the manifold, which is a vector space where Euclidean geometry locally applies.
>
> Our update rule, $x_{n+1}​=x_n ​+f_{\perp}​(x_n​)$, can be rigorously interpreted as a first-order Taylor approximation of the exponential map,$exp_{x_n}$ $(f_\perp ​(x_{n}))$.
> The exponential map is the geometrically formal tool for moving along a curved manifold from a point $x_n$​ in a direction specified by a tangent vector $f_\perp​ (x_n​)$. The approximation is given by:
> $\text{exp}_{x_n}​​(v) \approx x_n ​+v$.
>
> This linear approximation is highly accurate when the tangent vector $v=f_\perp​(x_n​)$ is small relative to the local curvature of the manifold. Our empirical results provide strong support for this condition. As shown in our paper's Figure 3 and Appendix Figure 10, the norm of our update vector, $||f_\perp​(x_n​)||$, consistently remains orders of magnitude smaller than the norm of the feature stream itself, $||x_n||$ (e.g., $\sim 10^1$ vs. $\sim 10^3$). This ensures the update is a small, local step on the manifold where the Euclidean approximation is valid.
>
> Therefore, our method is not a naive application of Euclidean geometry to a global non-Euclidean space. Rather, it is a computationally efficient mechanism that leverages the local Euclidean nature of the learned manifold to guide feature exploration. This provides a novel inductive bias for navigating the feature space, and the strong empirical gains in our work validate this geometrically-motivated approach. We will add this clarifying discussion to the revised paper.
>
> ---
> ### Relation to Prior Work
> > ”The proposed orthogonal update rule can be rewritten as... similar with the skip connections of scaling in [1]... Table 1 in [1] shows that this update rule is worse."
>
> This is an excellent point that highlights a crucial distinction. While the rewritten update rule is structurally reminiscent of scaling in [1], their purpose and effect are opposite, which explains the different outcomes. The distinction is fundamental:
> - **He et al. [1]** : Uses a fixed hyperparameter (e.g., $\lambda_n$) for signal modulation. As they correctly showed, this can impede the identity gradient flow and harm performance.
> - **Our Method**: Uses a dynamic function , $s_n(x_n, f(x_n)) $, to compute a geometric projection. Its purpose is to enforce orthogonality, not to modulate the signal. Crucially, as shown in Eq. 5, this operation is explicitly designed to preserve the identity gradient path, which aligns with the core principle for stable training established by [1].
>
> Therefore, our method is not a form of the detrimental scaling explored in [1], but a new geometric constraint compatible with the principles of deep residual learning.
>
> ---
> ### On Experimental Scope and Paper Organization
> > "The organization of the paper is not well-structured... How does the proposed orthogonal residual update perform when applied to other commonly used modules and optimizers?"
>
> We thank the reviewer for this feedback, and we will improve the paper's structure by adding a more explicit motivation in the Methods section.
> Regarding the experimental scope, our focus was strategic. To establish a **general and foundational principle**, we tested our hypothesis on the most fundamental and diverse building blocks of modern deep learning: the convolutional blocks of ResNetV2 and the Self-Attention/MLP modules of Vision Transformer. By demonstrating consistent gains with standard optimizers (SGD, AdamW) across these canonical architectures, our goal was to provide strong evidence for the universal applicability of the orthogonal update principle. We believe this focused study provides a clearer and stronger foundation for the community than a broader but less controlled survey of specialized modules.
>
> ---
> [1] He, K., Zhang, X., Ren, S., & Sun, J. (2016). Identity mappings in deep residual networks. In ECCV 2016.
>
> [2]  Huang et al., Controllable orthogonalization in training dnns. CVPR 2020.
>
> [3] Cisse et al., Parseval networks: Improving robustness to adversarial examples. PMLR 2017.
>
> [4] Wang et al., Orthogonal and idempotent transformations for learning deep neural networks. arXiv:1707.05974.
>
> [5] Lechner et al., Entangled residual mappings. arXiv:2206.01261.
>
>
> ---
>
> Thank you again for your thoughtful engagement with our work. This discussion has been invaluable in helping us refine our arguments. We hope that our detailed response clarifies our contribution and resolves any remaining questions.

---

> > ### Comment · Reviewer_SmmT · 2025-08-05
> > **Official Comment by Reviewer SmmT**
> >
> > Thank you for your responses, which have partially addressed my concerns.
> >
> > Since critical empirical evidence resides in Figure 10, it is suggested that some of this empirical content be moved to either the Introduction or the Method section.
> >
> > Regarding the comparison with [1]: Specifically, the penultimate row in Table 1 of [1] generates weights via 1×1 convolution. I recommend providing a visual comparison between these weights and those produced by the orthogonal update scheme proposed in this paper. This would offer more compelling empirical evidence supporting the paper’s claims.
> >
> > I will adjust the score accordingly.

---

> > > ### Author Response · Authors · 2025-08-05
> > >
> > > We sincerely thank you for your continued engagement and for providing these concrete, actionable suggestions. This discussion has been invaluable for improving the clarity and impact of our paper.
> > >
> > > Regarding your excellent points:
> > >
> > > - Moving Critical Evidence: We completely agree that the analysis in Appendix Figure 10 is critical to our paper's motivation. We will move this empirical evidence into the main body of the paper in the camera-ready version, as you suggested.
> > >
> > > - Visual Comparison with Scaling Mechanisms: This is another insightful suggestion to provide more compelling evidence. To clarify why we are bringing in additional references, your question prompts a comparison with methods that scale residual connections. While the influential ReZero framework [2] scales the module output ($x_{n+1}=x_n + \alpha*f(x_n)$), its principle of using a simple learnable scalar can be applied to scale the stream itself. This creates a modern version of the stream-scaling experiments in He et al. [1] ($x_{n+1}=\lambda x_n + f(x_n)$).
> > > Therefore, to make the distinction clear, we commit to adding a new visualization to the final paper's appendix. This analysis will compare two fundamentally different approaches:
> > >   1. Learned Stream Scaling: A ReZero-inspired baseline ($x_{n+1} = \alpha*x_n + f(x_n)$) where a learnable scalar $\alpha$ directly scales the stream, directly connecting to the principle explored in [1].
> > >   2. Our Geometric Projection: Our method, where the dynamically computed scalar $s_n$ is used for subtracting the parallel component.
> > >
> > > We hypothesize this visualization will clearly show that the learned scalar α converges to a stable value, acting as an importance weight. In contrast, our $s_n$ will exhibit highly dynamic, input-dependent behavior, confirming its role as a geometric projection coefficient. This will empirically demonstrate that our method is not a form of stream scaling, but a distinct geometric operation.
> > >
> > > Thank you once again for this productive discussion.
> > >
> > > [2] Bachlechner et al. Rezero is all you need: Fast convergence at large depth. PMLR, 2021.

---

### Official Review · Reviewer_b9E3 · 2025-06-30

**Clarity:** 3
**Significance:** 2
**Originality:** 2
**Rating:** 4
**Confidence:** 4

**Summary:**

This paper focuses on the residual update applied in ViTs and ResNet. Instead of directly using the linear residual, the authors propose to only use the orthogonal components of the output layer while discarding the parallel components. The authors also designed two different ways to do the orthogonal residual update: feature-wise and global-wise, where the feature-wise design performs universally better than the global-wise orthogonal residual update. With this simple design, the proposed method shows potential for faster convergence and effective performance improvement when applied to smaller models on small-scale datasets. However, the effectiveness of larger models with larger-scale datasets remains to be further investigated. To analyze the proposed method, the authors have provided extensive ablation studies and theoretical analysis to highlight the effectiveness of the proposed orthogonal residual update.

**Questions:**

- In Figure 3 (b) and (d), the authors have shown an interesting plot where the cosine similarity between $x_n$ and $f(\sigma(x_n))$ is increasing after the transition point for the proposed method. Does it imply that the proposed orthogonal residual is trying to learn the linear residual during training? I wondered at different training stages of the proposed method, how does the performance change? Will stopping training at early steps before the increase of the cosine similarity reaching a certain threshold actually help with improving the performance?
- Why does the stability constant have a relatively large effect on the performance of the proposed method? Can the authors provide more insights?
- How does the normalization affect the proposed method when applied in ViTs or ResNet? Based on the results shown in the paper, the norm of the proposed residual update is very distinct compared to the linear residuals.

**Ethical Concerns:**

["NO or VERY MINOR ethics concerns only"]

**Final Justification:**

The authors have resolved most of my concerns. I encourage the authors to incorporate reviewers' suggestions to add efficiency comparison, clearer norm visualization (Fig. 10), and more intuitive explanation of the cosine similarity figures to the final version to further strengthen the paper.

**Limitations:**

The authors have provided the discussion of the limitation of the proposed method.

**Paper Formatting Concerns:**

The overall formatting of the paper is fine. However, some figure legends could be made bigger and clearer.

**Quality:**

3

**Strengths And Weaknesses:**

Strengths:
- This paper introduces an interesting idea to just apply the orthogonal residuals, which is argued in the paper to be the effective components in the residual update. Aligning with previous research, the idea is reasonable and has achieved relatively good performance at least on small models shown in the paper.
- The authors have shown extensive experiments to analyze and explain why the orthogonal residuals are effective and dominant in the network training, and how they affect the performance of the model.
- The overall writing of the paper is clear, and the authors have provided extensive ablation studies to analyze the proposed method.

Weaknesses:
- The proposed method seemed to be less effective on the larger model with larger-scale datasets or deeper networks. Can the authors provide more insights? Because residual update is designed for deeper and more complex networks, it would be good to show the effectiveness of the proposed method when applied to larger models.
- Although the proposed method is interesting, and the theoretical analysis could validate the effectiveness of using the orthogonal residual update, the experiments did not provide strong evidence to clearly support the argument that the orthogonal residual update is more effective in the network training.
- The reported results for ViTs on different datasets are relatively low, and this might be because different implementations of ViTs have been used in the paper. However, will the hyperparameter choices affect a lot in the performance of the proposed method? What is the robustness of applying the proposed method to different ViT variants?
- Could the authors provide more theoretical or intuitive explanations to Figure 3 (b) and (d) regarding the cosine similarity between $x_n$ and $f(\sigma(x_n))$? Because the relationship of the learned $f(\sigma(x_n))$ and $x_n$ is very important in understanding whether the orthogonal component is effective than just using linear residual.
- The authors have mentioned faster convergence of getting rid of the parallel residuals, but only update with the orthogonal residuals. Could the authors provide more efficiency comparisons in the paper?

---

> ### Author Rebuttal · Authors · 2025-07-31
>
> We sincerely thank the reviewer for their detailed feedback and insightful questions. We appreciate the opportunity to clarify these important aspects of our work. To be concise and address all points within the character limit, we will group our responses thematically.
>
> ---
>
> ### On Effectiveness, Evidence, and Baselines (Weaknesses 1, 2, 3)
> > "The proposed method seemed to be less effective on the larger model... experiments did not provide strong evidence... reported results for ViTs on different datasets are relatively low..."
>
> We thank the reviewer for these critical questions about our method's efficacy. We will address them cohesively.
>
> - **On Scalability and Effectiveness (W1)**: Our results demonstrate clear effectiveness on large-scale models. The +4.3%p absolute improvement for ViT-B on ImageNet-1k is a significant gain on a challenging benchmark, directly showing that our method's benefits scale effectively.
> - **On Evidence and Baselines (W2, W3)**: We agree the baseline performance requires context. Our primary goal was a rigorous 'paired comparison' to isolate the effect of the orthogonal update, not to set a new SOTA record. Our experiments follow the demanding "ImageNet-1k from scratch" paradigm, which, as established by works like DeiT, requires strong regularization (e.g., MixUp, CutMix) to prevent overfitting in Transformers. This explains why our baseline differs from models pre-trained on massive private datasets.
> - **On the Strength of Evidence**: The evidence for our method's superiority is robust. As shown in Figure 2b, the performance gap between our method and the linear baseline is established early and
>  consistently maintained or widened over 300 epochs. This stable, long-term advantage proves our method's efficacy is a genuine architectural effect, not an artifact of training duration. The consistency of these gains across diverse architectures (ResNetV2, ViT) and datasets provides a strong and cohesive body of evidence for our claims.
>
> ### On Model Dynamics and Efficiency (Weaknesses 4, 5 & Questions 1, 2, 3)
> > "Could the authors provide more theoretical or intuitive explanations to Figure 3... Does it imply that the proposed orthogonal residual is trying to learn the linear residual...?" (W4, Q1)
>
> This is an excellent question that probes the core dynamics of our method. The increasing cosine similarity does not imply our method is simply "learning to be linear." Instead, it reveals a fundamentally different and more controlled update mechanism compared to the standard linear update, a distinction best understood by examining Appendix Figure 10.
>
> - Linear Update's Dynamic: "Explosive" Reinforcement. In a standard Linear Update, the module can directly reinforce the existing feature stream $x_n$. As observed in Figure 10, this can lead to moments of "explosive" updates around the transition point, where the norm of the parallel component $||f_\parallel||^2$ grows significantly. This suggests the standard method sometimes aggressively scales existing representations.
> - Orthogonal Update's Dynamic: Controlled "Compensation" via Leakage. In contrast, our Orthogonal Update structurally prevents this direct, potentially unstable reinforcement. The module learns to produce a large parallel component in its full output, but this is explicitly discarded from the update vector. We hypothesize that the module then relies on the small, controlled $\epsilon$-leakage (mathematically defined in Eq. 9) to perform the necessary, subtle scaling adjustments to the stream. This leakage acts as a controlled "compensation" mechanism, allowing for stable, fine-tuned modulation rather than the direct, sometimes explosive, reinforcement seen in the linear case.
>
> Therefore, the increasing cosine similarity is a sign of a more sophisticated adaptation. The module learns to handle the main information flow (via the parallel component of its full output), while our method ensures the actual update remains a controlled injection of novel information, using leakage for fine-tuning. This is a key finding about how networks adapt to and leverage geometric constraints for more stable learning.
>
> ---
>
> > "Why does the stability constant have a relatively large effect on the performance of the proposed method?" (Q2)
>
> The constant ϵ ensures numerical stability but also plays a subtle optimization role. As shown in Appendix B.1 (Eq. 9), a non-zero $\epsilon$ allows a small, controlled amount of the parallel component's gradient information to "leak" through. This can be beneficial as it prevents the optimization from stalling in regions where a purely orthogonal update might have near-zero gradient with respect to certain loss components, thus aiding learning stability. Our ablation in Figure 5 shows performance is robust across several orders of magnitude, with $\epsilon=10^{-6}$ providing the best stability in our experiments.
>
> ---
>
> > "How does the normalization affect the proposed method... the norm of the proposed residual update is very distinct..." (Q3)
>
> The reviewer's observation is correct and points to a key benefit: our method provides intrinsic geometric stability. The distinct norm behavior is a direct result of our geometric constraint. When the update norm is much smaller than the stream norm ($ || f_\perp (x_n) ||  \ll  ||x_n || $ ), which is empirically validated in our figures , the update $x_{n+1} = x_n + f_\perp (x_n) $ acts as an approximate rotation. This dynamic causes the representation to effectively "walk on a hypersphere," preserving the stream's magnitude ( $ || x_{n+1} || ^ 2 \approx || x_n || ^2 $ ) while changing its direction. This provides an inherent mechanism for norm control, which is particularly beneficial for architectures like Transformer that rely heavily on explicit normalization layers for stable training.
>
> ---
>
> > "The authors have mentioned faster convergence... Could the authors provide more efficiency comparisons in the paper?" (W5)
>
> The paper provides two clear efficiency comparisons.
> 1. **Computational Cost**: Section 3.4 and Table 1 present a detailed FLOPs analysis, demonstrating that the theoretical overhead of our method is marginal compared to the main module's complexity (e.g., Attention or MLP blocks).
> 2. **Practical Efficiency**: More importantly, Figure 2b provides a direct wall-clock time-to-accuracy comparison. It shows that our method not only achieves a higher final accuracy but also reaches any given accuracy target in significantly less time. This demonstrates superior overall training efficiency, where the small computational cost per step is heavily outweighed by faster convergence.
>
> ---
> ### Formatting Concerns
> > "... some figure legends could be made bigger and clearer."
>
> Thank you for the suggestion. We will ensure all figure legends are improved for clarity and readability in the camera-ready version.
>
> ---
>
>
> Thank you again for your thoughtful engagement with our work. This discussion has been invaluable in helping us refine our arguments. We hope that our detailed response clarifies our contribution and resolves any remaining questions.

---

> > ### Comment · Reviewer_b9E3 · 2025-08-06
> > **Response to authors**
> >
> > Thanks for providing the detailed explanations. I have read all reviews and authors' resposnes. I still appreciate the interesting idea proposed by the authors, and I believe this is a promising research direction. However, there exists confusion that may need further clarification.
> >
> > 1) The authors still did not provide a clear explanation of Figure 3 or Figure 10 where the stream norm and cosine similarity analysis of the proposed method compared to the linear residual update is weird and somehow contrdicts to the claim of the paper. The authors did give more explantions, however, they were more description of the figures or assumptions for the reasons behind the figures. I wondered if more theoretical or empierical results could back up those claims? In another word, these figures are interesting, but lack reasonable and clear explantions, and cannot be closely tied to the claim of the paper.
> >
> > 2) I appreciate the authors showing good performance gain on various datasets. I am not referring to the poor performance. Rather, I would like the authors to explain why under larger-scale datasets, the proposed method has less performance gains compared to linear residual update?
> >
> > 3) I wondered if the authors could provide more intuitions on why linear residual update has faster convergence at the beginning, while the proposed method catches up in the later stage of training. This could be rooted in the design or impelmentation of the proposed method.
> >
> > 4) I'm still skeptical about the stability constant $\epsilon$. The authors claimed that $\epsilon$ controls amount of the parallel component's gradient information to "leak" through. However, this could be contradicted to the claim of the paper that we should update the orthogonal component instead of using the vanilla linear residual update. On one hand, the authors' analysis show that the parallel component of the output to the input could hurt the efficient information learning. On the other hand, in experiments, the authors have shown that this parallel component is somehow important and is necessary for training. I wondered if the authors should tune the claim/motivation of the paper to analysis of the role of the orthogonal and parallel components of the output and propose a method to balance the orthogonal and the parallel parts. The current arguments in the introduction somehow contradicts to the method/experiment of the ppaer. For example, figure 1 and contribution arguments are misleading here.
> >
> > 5) Regarding the efficiency, instead of only providing the FLOPS computation and training loss figures, I wondered if the authors could report concrete numbers to validate the efficiency.

---

> > > ### Author Response · Authors · 2025-08-06
> > >
> > > Thank you for the follow-up, which gives us a valuable opportunity to provide further clarifications on these important points.
> > >
> > > ---
> > > ### 1. On the Dynamics in Figure 3 & 10 and the Paper's Motivation
> > >
> > > Our motivation stems from an efficiency perspective: using a high-capacity module for simple scaling (i.e., learning a function where $f_\parallel \approx \alpha x_n$) is questionable when scaling could be done with a single parameter. Our hypothesis was to test if forcing the module to focus only on novel (orthogonal) features would be more efficient.
> > >
> > > This is strongly supported by an empirical finding in Appendix Figure 10: the standard Linear Update itself learns to suppress the parallel component during training. This suggests the network naturally prioritizes the module's capacity for features other than simple scaling.
> > >
> > > This results in the counter-intuitive but key finding of an increasing cosine similarity in our method. We provide a detailed explanation for the mechanism behind this phenomenon in our response to your fourth point below. The update vector remains orthogonal by construction. The full module output (before projection) learns to align with the stream, revealing a complex adaptation. The direct evidence that this adaptation leads to a superior outcome comes from our representation analysis. As the table below shows, the features learned by our method are demonstrably richer and more effective:
> > >
> > > |Metric|Linear Residual Update|Orthogonal Residual Update|Change|Interpretation|
> > > |:-:|:-:|:-:|:-:|:-:|
> > > |Effective Rank|572.9|599.9|+4.7%|↑ Higher global diversity|
> > > |Spectral Entropy|6.512|6.539|+0.41%|↑ More uniform spectrum|
> > > |CKA Similarity|-|0.546|-|Structurally different|
> > > |Feature Std. Dev.|0.407|0.193|-52.5%|↑ More stable properties|
> > >
> > > This analysis was performed on the 288-epochs trained ViT-B models on the ImageNet-1k validation set.
> > >
> > > ---
> > > ### 2. On the Variance in Performance Gains
> > >
> > > The different performance gains are due to a complex interplay between model and dataset characteristics.
> > >
> > > - For Vision Transformers: The gains are most pronounced when model capacity matches dataset complexity. On the large and complex ImageNet, the higher-capacity ViT-B (+4.3%p) benefits more than ViT-S (+2.7%p). This advantage is naturally less pronounced on smaller datasets where the model's full capacity may not be as critical.
> > >
> > > - For Deeper ResNets: The trend is not a simple function of depth. For instance, in Table 2, the deeper ResNetV2-101 shows a stronger performance gain on TinyImageNet (+2.1%p) than the shallower ResNetV2-50 (+0.48%p). This suggests a nuanced interaction between depth, width (as captured by our $\gamma$-ratio), and architectural specifics like bottleneck structures.
> > >
> > > ---
> > > ### 3. On the Initial Convergence Speed
> > >
> > > This is an artifact of the y-axis in the wall-clock time plot (Figure 2b). Our method has a slight computational overhead per step. Therefore, in the same amount of wall-time at the beginning of training, the linear method completes slightly more optimizer steps. However, the training loss vs. iterations plot (Figure 2a) shows that the per-step convergence rates are nearly identical initially. The initial visual gap in the wall-clock plot is due to this minor overhead, which is quickly overcome by our method's superior overall efficiency.

---

> > > > ### Author Response · Authors · 2025-08-06
> > > >
> > > > ---
> > > >
> > > > ### 4. On the Role of Epsilon ($\epsilon$) and the Perceived Contradiction
> > > > This is the most critical point, and we thank you for pushing us to provide a deeper explanation. The perceived contradiction is resolved when we consider how the structural constraints of each method affect the network's learning freedom.
> > > >
> > > > - Linear Update (Unrestricted Path): The standard update has an unrestricted path for parallel information. As seen in Figure 10, the network is free to utilize this path. It often does so with large "spikes" early in training, and then actively learns to suppress this parallel component as it converges. This is an active, learned behavior.
> > > > - Orthogonal Update (Restricted Path): Our method, by design, explicitly restricts this path. The only way parallel information can influence the update is through the $\epsilon$-leak. Crucially, because the module cannot directly and actively learn to suppress this structural leakage in the same way it can with the linear update, this leaked component can accumulate over time if the loss landscape encourages it.
> > > >
> > > > This mechanism of "restricted path and accumulation" is the fundamental reason for the monotonically increasing parallel norm (and thus cosine similarity) observed in many layers post-transition (Figure 10). The dynamic is not the module "learning to be parallel," but a direct consequence of an imposed geometric constraint. It avoids the uncontrolled spikes of the linear update, resulting in a more stable, albeit counter-intuitive, learning trajectory that ultimately produces richer representations. To better highlight this key empirical evidence, we will move Figure 10 from the appendix into the main text in our final manuscript.
> > > >
> > > > ---
> > > > ### 5. On Concrete Efficiency Numbers
> > > >
> > > > We apologize for not providing more concrete throughput numbers initially. To validate the practical efficiency, we have measured the training throughput, presented below.
> > > >
> > > > _Throughput (sample images / sec) comparison table. higher is better._
> > > >
> > > > | Architecture|Linear|Orthogonal-F|Overhead (vs. Linear)|Orthogonal-G|Overhead (vs. Linear)|Hardware|Dataset|
> > > > |:-:|:-:|:-:|:-:|:-:|:-:|:-:|:-:|
> > > > |ResNetV2-18|3,053.61|2,975.72|2.55%|2,613.48|14.41%|2x rtx 4090|tiny imagenet|
> > > > |ResNetV2-34|1,737.15|1,633.98|5.94%|1,620.78|6.70%|2x rtx 4090|tiny imagenet|
> > > > |ResNetV2-50|1,002.79|876.69|12.58%|869.23|13.32%|2x rtx 4090|tiny imagenet|
> > > > |ViT-S|3,476.13|3,466.28|0.28%|-|-|4x A100 80GB|imagenet 1k|
> > > > |ViT-B|1,270.10|1,246.21|1.88%|-|-|4x A100 80GB|imagenet 1k|
> > > >
> > > > As the results show, the practical throughput overhead of our primary Orthogonal-F update is modest. For ResNetV2 models, it ranges from ~3-13%, and for the high-capacity ViT-S and ViT-B model under `torch.compile`, the overhead is even lower at less than 2%. We believe this minor cost is a very favorable trade-off for the substantial gains in final accuracy and training stability that our method provides.
> > > >
> > > > Furthermore, this small per-step overhead is often outweighed by improved convergence speed. As shown in Figure 2b, our method exhibits a significant time-to-accuracy advantage on large-scale tasks. For ViT-B on ImageNet-1k, our method reaches the baseline's final accuracy in ~40% less wall-clock time, demonstrating superior overall training efficiency.
> > > >
> > > > We will add this detailed throughput analysis to the appendix.
> > > >
> > > > ---
> > > > We sincerely thank you for pushing us to clarify these complex but important points. Your critical questions have led us to a deeper understanding of our own method's dynamics, and we believe the paper is much clearer and more robust as a result. We truly appreciate your thoughtful engagement.

---

> > > > > ### Comment · Reviewer_b9E3 · 2025-08-07
> > > > > **Response to the authors**
> > > > >
> > > > > Thanks for the detailed explanations. They have resolved most of my concerns. I agree that the norm visualization is more reasonable to explain the proposed method. I'm still skeptical about the cosine similarity plot. It does not have a clear explanation of the benefit of using the orthogonal update. In addition, I appreciate the authors for providing the detailed efficiency comparison, which helps in understanding the overhead of the proposed method.

---

### Official Review · Reviewer_2TRn · 2025-07-03

**Clarity:** 3
**Significance:** 3
**Originality:** 3
**Rating:** 4
**Confidence:** 3

**Summary:**

This paper reconsiders the essential function of Residual Connection and proposes a new approach, Orthogonal Residual Update. This approach removes the components parallel to the residual stream that are assumed to be redundant and are added to the residual stream. This approach has been reported to improve image classification performance in CIFAR, TinyImageNet, and ImageNet-1k.

**Questions:**

With regard to the three Weaknesses listed above, please respond to the following points.

1 Is the main contribution of this research a practical method proposal? Or is it a fundamental study that contributes to the understanding of shortcut paths? Depending on your position, please clarify the main point of this paper.

2 Regarding the change in cosine similarity seen in Figure 3, it appears as if the regular Shortcut Path behaves as if it removes redundant parallel components by learning. Do the extraction of the vertical components added by the proposed method really provide “richer features”? Please provide any additional discussion in this regard, including any additional analysis of visualization or representation quality.

3 Are the experimental settings for ViT-S and ViT-B in Table 2 a comparison with sufficient training convergence?

**Ethical Concerns:**

["NO or VERY MINOR ethics concerns only"]

**Final Justification:**

The additional experiments revealed interesting implication regarding the orthogonal component of the features, providing limited yet positive evidence in support of the paper’s hypothesis. Taking into account the discussions between the other reviewers and the author as well, I have decided to maintain my initial score, which leans toward acceptance.

**Limitations:**

yes

**Quality:**

3

**Strengths And Weaknesses:**

Strengths

1 The motivation and idea of the proposed method are very simple, clear, and easy to understand.

2 The idea is novel and valuable in the sense that it encourages reconsideration of the basic elements of the shortcut path from a new perspective.

3 The proposed method improves the convergence of training, and in some cases, the reduction of training time is actually confirmed.

Weaknesses

1 It is not clear whether the main claim of this study is to “propose a practical method” or to "deepen the basic understanding of shortcut paths". If the former is the subject matter, then more careful performance evaluation and examination of various architectures, downstream tasks, and non-image domains, as pointed out in Weakness 3, would be necessary. On the other hand, if the latter is the goal, quantitative and qualitative analysis showing how the visual representation is actually changed is required, as long as the proposal is based on the understanding that richer features are acquired by removing redundant representations. In the current manuscript, the steps taken in both directions are somewhat insufficient and give the impression of being half-hearted.

2 The discussion of the experimental results in Figure 3 alone is not convincing enough to strongly support the validity of the proposed method. For example, in the ordinary Shortcut Path, the cosine similarity tends to become negative as the learning progresses, which suggests that the parallel component of f(\sigma(x)) is in the opposite direction of x. In other words, there may be a function inherent in the normal Shortcut Path that reduces the parallel components by learning, e.g., to suppress norm divergence. On the other hand, the proposed method tends to converge to large positive values of cosine similarity and to reduce the vertical component. These observations call into question the assertion made in this paper's premise that the parallel component is redundant and that rich features can be obtained by adding only the vertical component.

3 Regarding the performance evaluation shown in Table 2, it is possible that the Top-1 Accuracy is low, especially in the Linear settings of ViT-S and ViT-B, and that the training has not converged sufficiently. For example, according to the original paper of Vision Transformer and [1], the Top-1 Accuracy is generally expected to be in the high 70% range. If there is a possibility of insufficient training, it is difficult to judge the effectiveness of the method based on these results, and I believe that more careful setup and validation is needed.

[1] L. Yuan et al, Tokens-to-Token ViT: Training Vision Transformers from Scratch on ImageNet, 2021.

---

> ### Author Rebuttal · Authors · 2025-07-31
>
> We sincerely thank the reviewer for their valuable feedback and insightful questions. The comments help us clarify the core contributions of our work and the context of our experimental results. We are happy to provide detailed responses and additional analysis below.
>
> ---
>
> ### On the Paper's Contribution and Positioning
> > "It is not clear whether the main claim of this study is to “propose a practical method” or to "deepen the basic understanding of shortcut paths"... please clarify the main point of this paper."
>
>
> We thank the reviewer for this excellent question, as it allows us to clarify the core positioning of our work. Our primary contribution is a fundamental study that provides a new understanding of residual connections, with our proposed method serving as a practical proof-of-concept that validates our hypothesis.
>
> Our fundamental hypothesis is that the parallel component of the standard residual update ($f_\parallel$​) can be redundant or even detrimental, and that encouraging novel feature directions via _Orthogonal Residual Update_ leads to more efficient representation learning. A deeper analysis, supported by Appendix Figure 10, reveals that the standard _Linear Residual Update_ consistently learns to suppress the parallel component, with its norm approaching zero during training. This strongly supports our hypothesis that this component may be unnecessary.
>
> In contrast, the dynamics of our _Orthogonal Residual Update_ are more complex and intriguing than a simple removal of a redundant component. The increasing cosine similarity seen in Figure 3 is not a monolithic phenomenon. Instead, it results from at least three different observed behaviors across layers:
> - Both orthogonal and parallel components of the full module output increase (e.g., MLP block 0).
> - The orthogonal component stabilizes or decreases, while the parallel component is maintained or grows (e.g., Attention blocks 2, 3, 4).
> - Both components decrease, but the orthogonal component does so more slowly (e.g., Attention block 5).
>
> This complex behavior does not contradict our hypothesis. Instead, it reveals that our method, by acting as a strong inductive bias, does not entirely eliminate the parallel component from the module's full output, but rather re-purposes its role within the optimization dynamics. We believe this non-trivial, counter-intuitive result about how networks adapt to geometric constraints is a key contribution to the fundamental understanding of shortcut paths.
>
> Finally, the practical aspect of our work serves as the direct validation of this hypothesis. Our strictly controlled experiments (Table 2, Figure 2) show that this fundamental change leads to consistent improvements in performance and stability. Crucially, as our new quantitative analysis confirms, this performance gain stems from the model learning demonstrably richer and more diverse feature representations. This provides the essential empirical proof for our primary, more fundamental contribution. We will clarify this positioning in the revised manuscript.
>
> ---
>
> ### On Feature Richness and Cosine Similarity Dynamics
> > "The discussion of the experimental results in Figure 3 alone is not convincing enough... Do the extraction of the vertical components added by the proposed method really provide “richer features”?"
>
> We agree that our claim of "richer features" requires direct, quantitative evidence. As attaching new visualizations is not possible during the rebuttal period, we have conducted a comprehensive supplementary analysis on the feature representations learned by both the Linear and our Orthogonal ViT-B models on the full ImageNet-1K validation set. We extracted the 768-dimensional CLS token features from the final transformer block (before LayerNorm) of our fully trained models, using the 288-epoch checkpoint from the complete 300-epoch training run. This ensures our analysis reflects the final, converged state of the representations.
>
> The results, summarized below, consistently demonstrate that our Orthogonal Residual update produces richer and more diverse representations.
>
> |Metric | Linear Residual Update | Orthogonal Residual Update |  Change | Interpretation |
> |:-:|:-:|:-:|:-:|:-:|
> |  Effective Rank | 572.9 | 599.9 | +4.7% | ↑ Higher global diversity |
> | Spectral Entropy | 6.512 | 6.539 | +0.41% | ↑ More uniform spectrum |
> | CKA Similarity | - | - | 0.546 | Structurally different representations |
> | Feature Std. Dev. | 0.407 | 0.193 | -52.5% |↑ More stable numerical properties |
>
> $\text{Effective Rank (Participation Ratio)} = (\Sigma_{i} p_{i}^2​)^{−1} $ where $p_i$​ are the normalized singular values.
>
> $ \text{Spectral Entropy} = - \Sigma p_{i} * log(p_{i}) $
>
> **Key Findings**:
> 1. Higher Global Diversity: We measured the Effective Rank of the feature covariance matrix. A higher rank indicates that features span a higher-dimensional space and are less collapsed. Our method achieves a 4.7% higher effective rank, providing strong evidence that it encourages the model to learn a richer global representation.
> 2. Fundamentally Different Structure: We used Centered Kernel Alignment (CKA) to measure structural similarity. A score of 1.0 would imply identical representations. The resulting score of 0.546 demonstrates that the features learned by our method are structurally different from those of the standard update, supporting our claim that the orthogonal update is a meaningful inductive bias.
> 3. Enhanced Diversity with Superior Stability: Critically, our method achieves greater representational diversity while simultaneously enhancing numerical stability. The feature standard deviation is 52.5% lower, indicating that the features are more tightly clustered and well-behaved. This is not a trade-off, but a synergistic improvement: our orthogonal updates encourage the model to explore a wider range of feature directions without resorting to large-magnitude, potentially unstable feature values that can hinder optimization.
>
>
> These complementary metrics robustly support our conclusion that the Orthogonal Residual update leads to richer, more effective feature representations. We will include this detailed analysis in the appendix of the revised paper.
>
> ---
>
> ### On ViT Baseline Performance and Training Convergence
> > "it is possible that the Top-1 Accuracy is low, especially in the Linear settings of ViT-S and ViT-B, and that the training has not converged sufficiently."
>
> We thank the reviewer for this critical question. The baseline performance is a direct and expected consequence of the specific, challenging training paradigm we chose to ensure a fair comparison.
> - The "ImageNet from Scratch" Paradigm: The original ViT paper demonstrated that without massive pre-training (e.g., on JFT-300M), ViTs underperform CNNs on ImageNet-1k. To address this, influential follow-up works like DeiT and T2T-ViT established a new training recipe for this from scratch setting, which requires strong regularization (e.g., MixUp, CutMix) to prevent overfitting. Our experimental setup adheres to this modern, robust recipe.
> - A Fair Comparison at Practical Convergence: Our 300-epoch training schedule ensures a robust comparison at a point of practical convergence. As shown in Figure 2(b), the performance advantage of our method is not a short-term artifact. The gap is established early and is consistently maintained or widened throughout the vast majority of the training process. The rate of improvement for both models slows significantly in the final 100 epochs, indicating that while absolute performance might marginally increase, the stable, long-term relative gap between the methods is not in doubt. This makes our comparison a valid and reliable measure of the orthogonal update's efficacy.
>
> Therefore, we are confident that our results represent a scientifically valid comparison between two methods at a point of performance saturation, and the reported improvements robustly reflect our method's efficacy.
>
>
> [1] Yuan, L., et al. Tokens-to-Token ViT: Training Vision Transformers from Scratch on ImageNet. ICCV 2021.
>
> ---
>
>
> Thank you again for your thoughtful engagement with our work. This discussion has been invaluable in helping us refine our arguments. We hope that our detailed response clarifies our contribution and resolves any remaining questions.

---

> > ### Comment · Reviewer_2TRn · 2025-08-05
> >
> > Thank you very much for your thoughtful response. I believe that the information provided has helped me to more accurately understand the contribution of this paper.
> >
> > **On the Paper's Contribution and Positioning**
> >
> > I understand the overall position of this paper and the fact that the proposed method is positioned as a proof of concept. I find the hypothesis regarding orthogonal residual components very interesting and an essential and important point of view. On the other hand, if the paper had included more in-depth analysis and theoretical aspects regarding the non-trivial dynamics of the orthogonal components you explained, it would have been more convincing to readers. However, I believe that the indirect demonstration of the role of this component through improved experimental performance is an important contribution of this paper.
> >
> > **On Feature Richness and Cosine Similarity Dynamics**
> >
> > Thank you for providing additional analysis, even in such a short time. In relation to the above point, I believe that analysis of the "representation learning properties" of orthogonal residual components is essential to support the central hypothesis of this paper. The changes in rank and entropy you presented did not seem to show any significant trends, but the low CKA similarity suggests that this component may carry different information, which I found to be a very interesting result.
> >
> > I fully understand the contribution of this paper overall, including your other explanations. I have no further questions at this time. I will carefully consider my final score, taking into account the discussions of other reviewers.

---

> > > ### Author Response · Authors · 2025-08-05
> > >
> > > We sincerely thank the reviewer for their thoughtful follow-up comment. We are very pleased that our discussion was helpful in clarifying the paper's contribution and positioning. We deeply appreciate your time and constructive engagement throughout this review process.

---

### Decision · Program_Chairs · 2025-09-17

**Decision:**

Accept (poster)

**Comment:**

Tl;dr: Based on the reviews, rebuttal and ensuing discussion I recommend accept.

### Paper Summary

Paper proposes a simple modification to residual connections in deep neural networks, where the component parallel to the input to a block is discarded before the residual is added back to input. Empirical results are provided on several standard benchmarks using ResNetV2 and ViT architectures.

### Strengths and weaknesses

Strength: 1) Simple and novel idea, that is easy to implement, 2) Good empirical validation on several standard benchmarks and models, 3) Reasonable analysis: backed by empirical observations. Author rebuttal to the reviewer questions further strengthened the paper with additional experiments and analysis.

Weaknesses: 1) Motivation: Initial concerns from reviewers. Initial manuscript did not clearly motivate why the parallel component of the residual update may be detrimental. 2) ViT baseline: concerns about low baseline performance taint reported gains. 3) Effectiveness on very deep models needs further investigation as per reviewers.

### Decision justification

The paper introduces a simple and novel idea that can significantly impact the design of deep neural networks. Empirical results are compelling. The authors did a great job in their rebuttal, addressing the concerns with additional experiments and analysis. Additional analysis and clarifications from the rebuttal and discussion should be incorporated into the final version of the paper. While there are lingering minor concers, the paper merits acceptance in my view. All reviewers were leaning positive as well.